# Dual contribution of ASIC1a channels in the spinal processing of pain information by deep projection neurons revealed by computational modeling

Magda Chafaï[1], Ariane Delrocq[1,2], Perrine Inquimbert[3], Ludivine Pidoux[1], Kevin Delanoe[1], Maurizio Toft[1], Frederic Brau[1], Eric Lingueglia[1], Romain Veltz[2]*, Emmanuel Deval[1]*

**1** Université Côte d'Azur, CNRS, IPMC, LabEx ICST, FHU InovPain, France, **2** Inria Center of University Côte d'Azur, France, Valbonne, France, **3** Université de Strasbourg, CNRS, Institut des Neurosciences Cellulaires et Intégratives, Strasbourg, France

* romain.veltz@inria.fr (RV), deval@ipmc.cnrs.fr (ED)

**Data Availability Statement:** The experimental data are available in open access in Zenodo (https://zenodo.org/record/7657384). The code is

## Abstract

Dorsal horn of the spinal cord is an important crossroad of pain neuraxis, especially for the neuronal plasticity mechanisms that can lead to chronic pain states. Windup is a well-known spinal pain facilitation process initially described several decades ago, but its exact mechanism is still not fully understood. Here, we combine both *ex vivo* and *in vivo* electrophysiological recordings of rat spinal neurons with computational modeling to demonstrate a role for ASIC1a-containing channels in the windup process. Spinal application of the ASIC1a inhibitory venom peptides mambalgin-1 and psalmotoxin-1 (PcTx1) significantly reduces the ability of deep wide dynamic range (WDR) neurons to develop windup *in vivo*. All deep WDR-like neurons recorded from spinal slices exhibit an ASIC current with biophysical and pharmacological characteristics consistent with functional expression of ASIC1a homomeric channels. A computational model of WDR neuron supplemented with different ASIC1a channel parameters accurately reproduces the experimental data, further supporting a positive contribution of these channels to windup. It also predicts a calcium-dependent windup decrease for elevated ASIC conductances, a phenomenon that was experimentally validated using the Texas coral snake ASIC-activating toxin (MitTx) and calcium-activated potassium channel inhibitory peptides (apamin and iberiotoxin). This study supports a dual contribution to windup of calcium permeable ASIC1a channels in deep laminae projecting neurons, promoting it upon moderate channel activity, but ultimately leading to calcium-dependent windup inhibition associated to potassium channels when activity increases.

available in ModelDB (http://modeldb.yale.edu/267666).

**Funding:** E.D. and E.L. received grants from the Agence Nationale de la Recherche, ANR (ANR-13-BSV4-0009 and ANR-17-CE16-0018), https://anr.fr/, and from the LabEx ICST (ANR-11-LABX-0015-01). E.D. received grants from the Association Française contre les Myopathies, AFM (grant #19618 and grant#23731), http://www.afm-telethon.com/, and from the Institut Analgesia/SFETD. E.D. and R.V. received a grant from the Neuromod Institute of the University Côte d'Azur (UCA), https://neuromod.univ-cotedazur.eu/. E.D. and L.P. have been supported by the French government, through the UCAJEDI Investments in the Future project managed by the National Research Agency (ANR) with the reference number ANR-15-IDEX-01. The funders had no role in study design, data collection and analysis, decision to publish, or preparation of the manuscript.

**Competing interests:** The authors have declared that no competing interests exist.

## Author summary

This work combines both *ex vivo* and *in vivo* electrophysiology to pharmacological approaches with animal toxins and computational modeling to report an unexpected dual participation of Acid-Sensing Ion Channels (ASICs) to the spinal pain facilitation process called windup. We demonstrate a functional expression of particular calcium permeable ASIC subtypes in spinal neurons associated to windup in deep dorsal horn laminae, and show that the *in vivo* spinal application of ASIC inhibitory toxins significantly reduces windup. A computational model of these neurons that includes ASIC and synaptic acidification parameters, reproduces experimental data with a windup increase when ASICs are progressively added to the model. But the model also unexpectedly predicts a windup decrease for high ASIC conductances, which has been experimentally validated *in vivo* using a potent pharmacological ASIC activator. Our experimental and computational data therefore support a bell-shaped modulation of spinal windup by ASICs, with a positive contribution for low to moderate activity but a negative calcium-dependent impact when their activity is strongly increased.

## Introduction

Dorsal horn of the spinal cord is a key point of the pain neuraxis where sensory-nociceptive information, coming from the periphery, enters the central nervous system to be integrated, processed, and sent to the brain. It consists of an extremely complex neuronal network organized in different laminae (laminae I to VI), including different types of projection neurons as well as excitatory/inhibitory interneurons (for review, see [1]). The complexity of dorsal spinal cord neuronal network not only lies on its architecture and its great neuronal diversity, but also on the fact it receives various sensory and nociceptive inputs from the periphery, and that it is modulated by descending pathways from supra-spinal levels. Spinal inputs come from peripheral Aβ, Aδ and C fibers and, importantly, these inputs can be subject to different facilitation/sensitization processes, leading to pain hypersensitivity and allodynia (for reviews, see [2,3]). These processes generally result from intense and repetitive noxious inputs and are associated with neuronal plasticity that sensitizes spinal neurons by increasing their spontaneous activity, decreasing their activation threshold, amplifying their response to stimuli and/or enlarging their receptive fields. Sensitized states can be long-lasting but are normally not permanent. However, they can also be associated to chronic pain states of clinical relevance [4], when pain loses its physiological protective function to fall into pathology.

Much progress has been made over the last decades in the understanding of spinal facilitation/sensitization molecular mechanisms, including windup, which is a "short term" facilitation process typical of wide dynamic range (WDR) projecting neurons [5]. Windup is a homosynaptic facilitation process of C-fiber inputs following peripheral low-frequency repetitive stimulations, resulting in a progressive increase of the number of action potentials (APs) evoked by WDR neurons [2]. Although windup and central sensitization share common properties, they are not equivalent but windup can lead to some aspects of central sensitization [6]. Therefore, windup remains an interesting way to study the processing of nociceptive information by spinal cord neurons (for reviews see, [2,7]). Many factors have been reported to contribute to and/or to modulate windup, among which the most important appear to be NMDA receptors [8,9], neurokinin-1 (NK1) receptors [10] and L-type calcium channels [11–13].

ASICs (Acid-Sensing Ion channels) are voltage-independent cationic channels mainly selective for Na$^+$ ions [14]. These channels are gated by protons, *i.e.*, they are sensors of

extracellular pH. Several ASIC subunits have been identified in mammals (ASIC1 to ASIC4, for reviews see [15,16]), which assemble as trimers [17] to form functional channels, including homomers and heteromers [18]. ASICs are widely expressed in the nervous system and are found throughout the pain neuraxis, including in spinal neurons of the dorsal horn [19–21]. The discovery and *in vivo* use of pharmacological tools able to modulate their activities (for review see [22]) have largely contributed to a body of proofs arguing for the involvement of these channels in pain, both in humans [23,24], and mainly in animal models of pain, either at the peripheral or the central level [25–30]. Interestingly, peptides isolated from animal venoms, which remain so far the most specific pharmacological tools able to modulate particular subtypes of ASICs [28,31–33], were reported to have strong analgesic or painful effects in animals [25–28,31,34,35], depending on whether they inhibit or activate the channels, respectively. ASICs are thus interesting pharmacological targets for pain, but if their role in peripheral sensory neurons is relatively well documented, little is known about their participation in pain processes in the central nervous system. For instance, while pharmacological inhibition of ASICs at the spinal cord level is known to produce potent analgesia [20,28,34], the mechanism of this effect still remains poorly understood.

In this work, we investigated the role of spinal ASICs, which are mainly formed by ASIC1a-containing channels [19–21], in the processing of pain information. By combining both *in vivo* and *ex vivo* neurophysiological approaches with pharmacology and computational modeling, we propose that ASIC1a homomeric channels in deep laminae projecting neurons participate to the spinal windup facilitation process with dual promoting and inhibitory contributions depending on their activity.

## Results

### Pharmacological inhibition of ASIC1a-type channels in the dorsal spinal cord decreases the evoked activity of WDR neurons

To study the *in vivo* role of spinal ASICs in the processing of sensory and nociceptive inputs, extracellular recordings of dorsal horn neurons (DH neurons) were performed on anesthetized rats (Figs 1 and S1 and S2). ASIC1a homomeric and ASIC1a/ASIC2 heteromeric channels have been reported to be predominant in dorsal spinal cord neurons [19–21]. *In vivo* recordings of DH neurons were thus combined with the use of mambalgin-1 and PcTx1, two inhibitory peptides specific for ASIC1-type channels, including ASIC1a homomers [28,33] and ASIC1a/ASIC2 heteromers [28,36–38], with nanomolar to few hundred nanomolar affinities. The combination of two independent selective inhibitors both increased the specificity of our results and allowed to gain information on the ASIC1a channel subtypes involved because of their partially overlapping pharmacological profiles. WDR neurons were classically identified according to their ability to evoke action potential (AP) firing in response to both innocuous (brush) and noxious (pinch) mechanical stimulations of their receptive fields on rat hind-paws (Fig 1A). A windup protocol was designed (16 repetitive electrical stimulations at 1 Hz) and applied onto the receptive fields of WDR neurons every 10 minutes, before (control) and after spinal application of mambalgin-1 (Fig 1B–1E) or PcTx1 (Fig 1F–1H). In control conditions, the number of APs evoked by C-fibers increased, as expected, with the number of stimulations, and the maximal windup was reached between the 13th and the 16th stimulation (Fig 1C and 1F). Spinal application of mambalgin-1 for 10 min drastically reduced the C-fiber induced windup by approximately 50% (Fig 1C), *i.e.*, 44% and 55% depending on whether the percentage of inhibition was calculated from the total number of AP (Fig 1D), or from the area under curve (Fig 1E), respectively. Interestingly, it also slightly but significantly decreased by 21% the number of APs at the input, which corresponds to the number of APs evoked by WDR

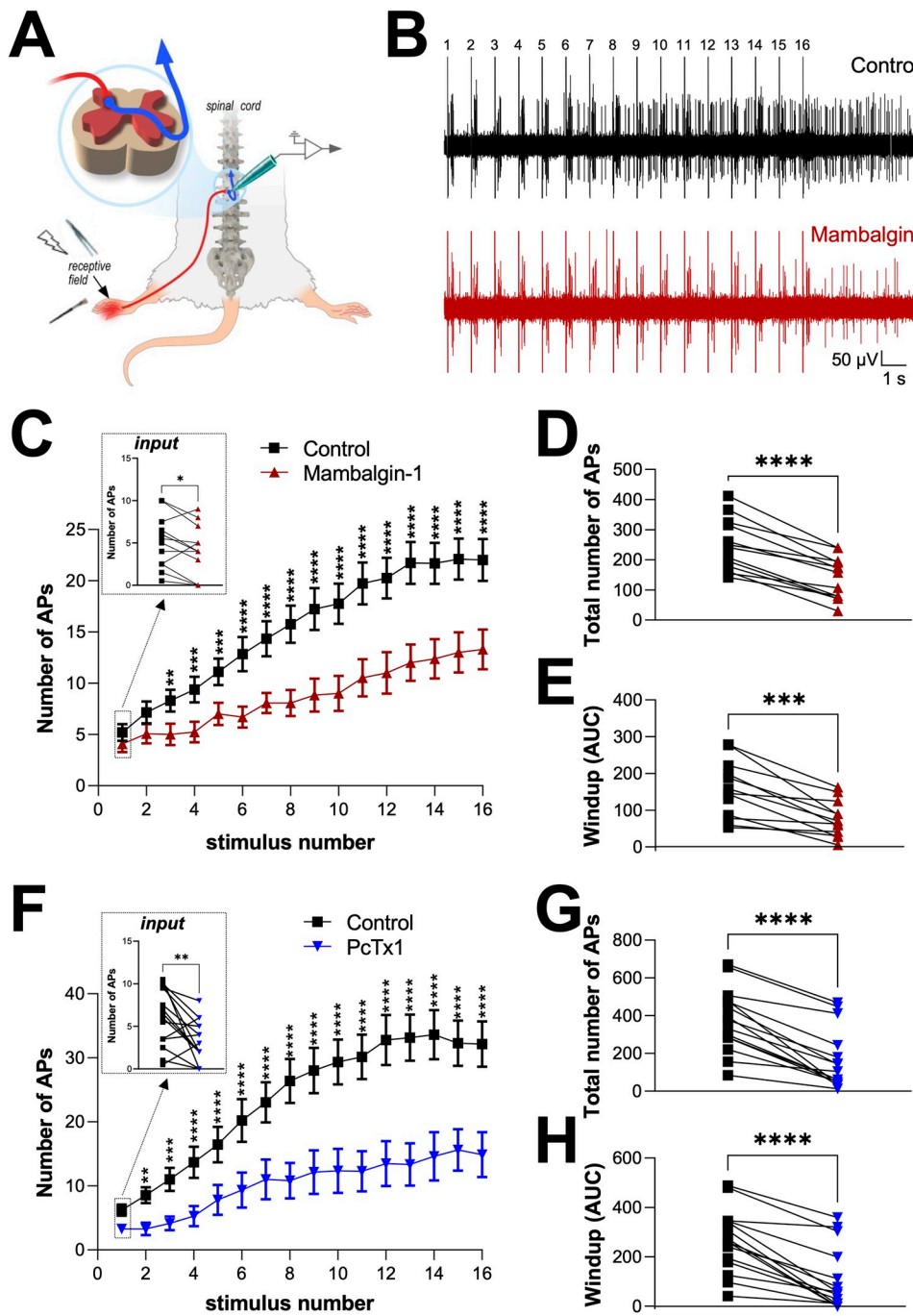

**Fig 1. Windup inhibition induced by spinal application of ASIC1a channel blockers.** *A*, Schematic representation of the *in vivo* electrophysiological method used to record WDR neurons from anesthetized rat. Receptive fields of neurons are stimulated either mechanically (brush, pinch) or electrically. *B*, Typical electrophysiological recording of a WDR neuron obtained following 16 electrical repetitive stimulations of its receptive field at 1Hz (windup protocol, see methods), before (control, top trace) and after spinal application of 30μM mambalgin-1 (bottom trace). *C*, Windup curves representing the number of C fiber-evoked APs for each of the 16 repetitive electrical stimulations in basal conditions (control, 1st), and after 10 min spinal application of mambalgin-1 (2nd, n = 13, **, *** and **** significantly different with $p < 0.01$, $p < 0.001$ and $p < 0.0001$, respectively, two-way ANOVA followed by a Sidak multiple comparison test). The *inset* shows the average number of APs evoked by the first electrical stimulation, *i.e.*, before establishment of windup (n = 13, *$p < 0.05$, paired t test). *D and E*, Statistical analyses of the total number of AP evoked by C fibers during a windup protocol (D), and of the global windup estimated as the area under curves (AUC, E) from the data shown C (n = 13, ***$p < 0.001$ and ****$p < 0.0001$, paired t tests). *F*, Windup curves representing the number of

C fiber-evoked AP for each of the 16 repetitive electrical stimulations in control (1$^{st}$) condition and after spinal application of 30μM PcTx1 (2$^{nd}$, n = 15, **, *** and **** significantly different with $p<0.01$, $p<0.001$ and $p<0.0001$, respectively, two-way ANOVA followed by a Sidak multiple comparison test). The *inset* shows the average number of AP evoked by the first electrical stimulation (n = 15, **$p<0.01$, paired t test). **G and H**, Statistical analyses of the total number of AP evoked by C fibers during a windup protocol (G), and of the global windup estimated as the area under curves (AUC, H) from the data shown F (n = 15, ****$p<0.0001$, paired t tests).

neurons at the first stimulation (Fig 1C, *inset*). The same kind of effects was also obtained when PcTx1 was delivered at the spinal level (Fig 1F–1H). Indeed, windup was inhibited by 57% or 60% in terms of total number of AP or AUC, respectively (Fig 1G and 1H). Moreover, PcTx1 also affected the input, with a 47% decrease of the number of AP induced at the first stimulation (Fig 1F, inset).

Spinal application of mambalgin-1 or PcTx1 also affected the Aβ- and Aδ-evoked activities in WDR neurons (S1 Fig), with small inhibitions of associated firings. Indeed, the firing induced by brushing (S1A Fig), which can be related to Aβ-fibers, was decreased by 20% in the presence of mambalgin-1, while PcTx1 had no effect (S1B Fig, $p = 0.07$ and $p = 0.31$, respectively). Moreover, both mambalgin-1 and PcTx1 significantly reduced the total number of AP induced by Aδ-fibers during the windup protocol (by 26% and 44%, respectively, S1C and S1D Fig).

All together, these results obtained with two independent selective inhibitors of ASIC1a channel subtypes demonstrate that spinal ASIC1a contribute to the integration of C-fiber inputs and associated windup. In addition, an involvement of either ASIC1a homomers or ASIC1a/ASIC2 heteromers is suggested from the partially overlapping pharmacological profiles of mambalgin-1 and PcTx1, which strongly affect ASIC1a homomeric channels [28,33] and have also been reported to be able to inhibit ASIC1a/ASIC2 heteromers [28,36–38] under specific conditions for PcTx1 [36–38].

## ASIC1a channel subtypes are functionally expressed in large WDR-like neurons from deep laminae V

To further investigate the contribution of spinal ASIC channels to WDR neuron activity, patch-clamp experiments were performed on spinal cord slices (Fig 2). Electrophysiological recordings were made in deep laminae V to record and characterize native ASIC currents in large neurons, which most likely correspond to the WDR neurons [39] studied *in vivo* (Fig 1). All the neurons recorded displayed an ASIC-type current in response to extracellular acidification from pH7.3 to pH6.6 (Fig 2A), with an average amplitude of 242 ± 37pA and an inactivation time constant of 1,536 ± 98ms (Fig 2B, left panel). Such inactivation kinetics were in the range of those of homomeric ASIC1a (Fig 2B, right panel) and heteromeric ASIC1a/ASIC2 [18,26] channels. On the other hand, the native ASIC current had a stable amplitude over time and no tachyphylaxis phenomenon [36,40] was observed upon repetitive activation at pH6.6 (Fig 2C), which was different from what has been classically described for ASIC1a homomeric channels [36,40], and to what was observed here in HEK293 cells transfected with ASIC1a (Fig 2C, inset). Tachyphylaxis has been reported to be absent in ASIC1a heteromeric channels containing the ASIC2 subunit [36,40], which would suggest that the native ASIC currents recorded from deep laminae neurons could be carried by ASIC1a/ASIC2 heteromers. However, native ASIC currents were almost completely blocked by extracellular applications of mambalgin-1 (1μM) and PcTx1 (30nM), although the conditions used are not supposed to lead to heteromer inhibition by the PcTx1 toxin [36,37] (Fig 2A and 2D).

To further investigate the effects of PcTx1 on homomeric ASIC1a and heteromeric ASIC1a/ASIC2 channels, HEK293 cells were transfected with either ASIC1a alone or ASIC1a

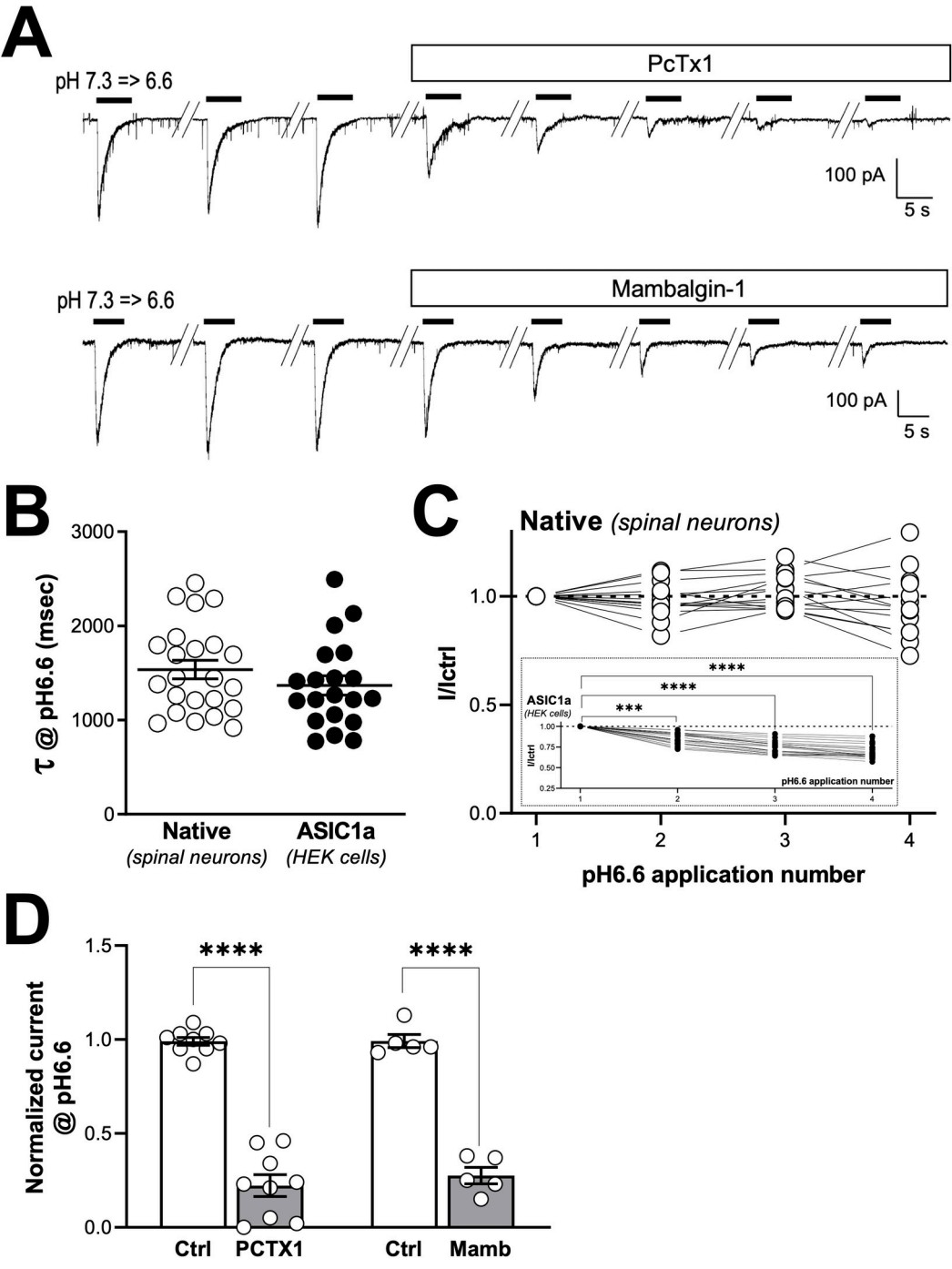

**Fig 2. Characterization of a native ASIC1a-type current in deep projection neurons from laminae V.** *A*, Typical voltage-clamp recordings of deep projection neurons (laminae V) obtained from spinal cord slices following extracellular acidifications from pH7.3 to pH6.6. Extracellular applications of the ASIC1a inhibitory peptides PcTx1 (30nM) or mambalgin-1 (1μM) both decrease the amplitude of native pH6.6-induced currents. *B*, Inactivation kinetics of native pH6.6-induced currents recorded in spinal neurons (native) are compared to those of ASIC1a homomeric currents form HEK293 transfected cells (form pH7.4 to 6.6). Inactivation rates (τ) were obtained by fitting current inactivation decays with a mono-exponential (n = 23 and 20 for spinal neurons and ASIC1a-HEK293 cells, respectively, *p* = 0.2399, unpaired *t*-test). *C*, Peak amplitude of the native ASIC current recorded in neurons following four consecutive pH6.6 extracellular acidifications (from 7.3 to 6.6 every 60s). Amplitudes are normalized to the first pH6.6-evoked current (n = 14, no significant tachyphylaxis process with *p* = 0.1882, one-way ANOVA test followed by a Dunnet's *post hoc* test). *Inset* shows the same experiment performed on ASIC1a homomeric current recorded in HEK293 cells (from 7.4 to 6.6 every 60s, n = 20,

significant tachyphylaxis process with ***$p<0.001$ and ****$p<0.0001$, one-way ANOVA test followed by a Dunnet's *post hoc* test). ***D***, Statistical analysis of both PcTx1 (30nM) and mambalgin-1 (1μM) effects on the native pH6.6-induced current amplitudes recorded as in A (n = 9 and 5, respectively, ****$p<0.0001$, paired Student t-test).

in combination with ASIC2a or ASIC2b subunits (Fig 3). The three different channel subtypes were clearly distinguishable according to their pH5.0-evoked currents, with a significant increase of the sustained/transient current ratio for ASIC1a/ASIC2a and ASIC1a/ASIC2b heteromers compared to ASIC1a homomeric channels (Fig 3A and 3B).The pH-dependence for activation was also different between the three ASIC1a channel subtypes, with a lower sensitivity for hereromers compared to the homomer as indicated by the pH6.6/pH5.0 current ratio (S2A Fig). Importantly, we found that ASIC1a homomer was the only ASIC1a channel subtype to be inhibited by PcTx1 in our conditions (Fig 3C), while mambalgin-1 inhibited both homomeric ASIC1a and heteromeric ASIC1a/ASIC2 channels as previously described [28] (Fig 3D).

Altogether, these data demonstrate for the first time that ASIC1a channel subtypes are functionally expressed in deep WDR neurons from laminae V, which is fully consistent with their participation to C-fiber input integration and windup. The properties of native ASIC1a-type currents, *i.e.* kinetics and more importantly pharmacological inhibition by both mambalgin-1 and PcTx1, suggest the expression of ASIC1a homomers rather than heteromers (*i.e.*, ASIC1a/ASIC2 channels) in these neurons.

## Computational model of WDR neuron including ASIC1a channel parameters and acidification of the synaptic cleft

A mathematical model of WDR neurons and windup was initially described by Aguiar and colleagues [41], and was later taken over by Radwani and colleagues [13] to help demonstrate the involvement of Cav1.3 channels in the genesis of windup. We decided to take advantage of this model to help us understand the role of ASIC1a channels in the processing of C-fiber input and windup in WDR neurons (Figs 4 and S3). In this model, a WDR neuron receives a direct input from an Aδ-fiber as well as an input from a C-fiber through an interneuron (S3A Fig). With this model, we were able to fully reproduce the windup described by Aguiar and colleagues [41], including the windup inhibition when NK1 ion channel parameters were modified (S3B Fig). The C-fiber-induced spiking activity predicted by the WDR neuron model was then compared to our experimental results, *i.e.*, an average of about 30 neurons recorded in control conditions. The spike time plot (S3C and S3D Fig) indicated that the model fitted better to experimental data when the interneuron between C-fibers and WDR neuron was removed. We thus decided to use a simplified model without interneuron (Fig 4A) to further study the involvement of ASIC1a in the windup process.

The WDR model was supplemented with either native ASIC1a homomeric or ASIC1a heteromeric channel parameters based on a recently described ASIC1a model [42] and previous data obtained in cultured rat spinal cord neurons [19] (see material and methods). A small calcium conductance was included to account for reported ASIC-mediated intracellular calcium increase ([14] and S2B Fig). Furthermore, as ASICs are proton-gated ion channels, we also needed to introduce a model of synaptic cleft acidification in our WDR neuron model. This particular point was achieved according to the work of Highstein and colleagues [43], where the synaptic pH is controlled by *(i)* a buffer, *(ii)* a homeostatic system that tends to bring the pH to the physiological value of 7.4 and, *(iii)* a proton current entering the synapse that generates acidification during synaptic activity (see material and methods). Using this model of synaptic pH control, the predicted acidification during a windup protocol is described in Fig 4B.

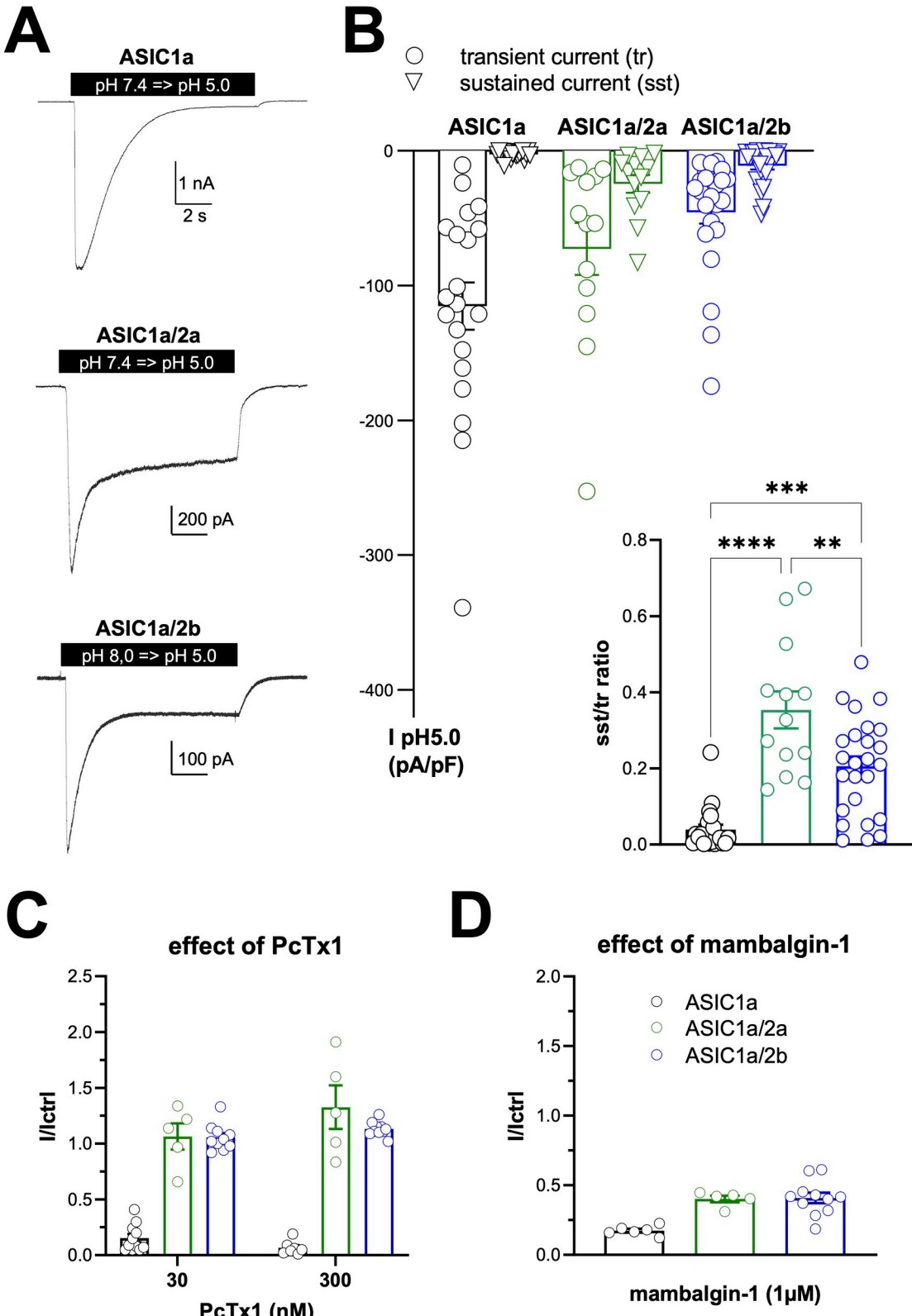

**Fig 3. Effects of PcTx1 and mambalgin-1 on ASIC1a homomeric and heteromeric channels. *A*,** Typical ASIC1a, ASIC1a/2a and ASIC1a/2b currents recorded at -50 mV from HEK293 transfected cells following extracellular acidification to pH5.0. ***B*,** Statistical analysis of the current densities recorded in A. Both the transient (tr) and sustained (sst) currents were measured, and the ratio sst/

tr is represented on the bargraph on the bottom right (n = 8–25, One-way ANOVA with $p < 0.0001$ followed by a Tukey's multiple comparison test, **** $p < 0.0001$). **C-D**, Effects of PcTx1 (30 nM or 300 nM) and mambalgin-1 (1μM) on pH-evoked ASIC1a (pH7.4 to pH6.0), ASIC1a/2a (pH7.4 to pH6.0) and ASIC1a/2b (pH8.0 to pH6.0) currents (n = 5–11).

Including native ASIC1a homomeric channel parameters in our WDR neuron model progressively increased the windup as the total ASIC conductance increased to 0.2nS (Fig 4C), in good agreement with experimental data showing a windup decrease following spinal application of mambalgin-1 or PcTx1 (Fig 1). However, when further increasing the ASIC1a conductance up to 1.4nS, the model showed progressive suppression of the windup (Fig 4D). We hypothesized that this effect was due to the ASIC-mediated increase of intracellular calcium concentration and subsequent activation of the hyperpolarizing calcium-dependent $K^+$ channels present in the model (S7 Fig). This was supported by the fact that removing the $Ca^{2+}$ permeability of ASICs restored the windup increase (Fig 4D).

Including native ASIC1a heteromeric parameters in our WDR neuron model gave qualitatively similar results, although within higher conductance range: a windup potentiation when ASIC conductance progressively increased up to 3nS, and a calcium-dependent windup decrease associated to higher ASIC conductances (Fig 4E and 4F). The ASIC conductance levels used in our models were in the range of the native ASIC1a-type global conductance estimated in WDR neurons (Fig 2), assuming the mean current amplitude recorded at -80 mV and a classical reversal potential of +50 mV (~ 2nS).

The mathematical model described here was thus fully consistent with an involvement in the windup process of homomeric or heteromeric ASIC1a-containing channels located in WDR neurons at the post-synapse with C-fibers. It also suggested a "bell-shape" participation of calcium-permeable ASIC1a to windup, with both positive and negative effects associated to low/medium and high conductances, respectively.

## Maximal activation of spinal ASIC1a channel subtypes inhibits windup through calcium-dependent $K^+$ channels

Our mathematical model predicted windup inhibition for high conductances of ASIC1a channels in WDR neurons, involving ASIC1a-associated calcium entry and likely subsequent activation of calcium-dependent $K^+$ channels. This hypothesis was tested experimentally by using MitTx (Fig 5), a peptide toxin with potent activating effects on ASICs and particularly ASIC1a channels [31]. MitTx was indeed reported to generate a maximal and persistent activation of the channels as well as a potent calcium response in neurons, which was abolished in ASIC1a knockout mice. Spinal application of MitTx induced a dose-dependent inhibition of windup (Fig 5A and 5B), in agreement with the effect predicted by our computational model. This effect started at $10^{-7}$M / $5.10^{-7}$M and reached a maximum at $10^{-6}$M. Inhibition of windup by MitTx was partially reversible upon washout (up to ~50% of control after 40 min of washout, Fig 1B). The computational model was next configured to mimic the experimental effect of MitTx, *i.e.*, with a sustained activation of ASIC channels at medium conductance, rather than very high conductances with normal dynamics (Fig 5C). This simulation confirmed that MitTx inhibits windup, in good agreement with experimental data.

To test the hypothesis that windup inhibition induced by MitTx was due to the opening of calcium-activated $K^+$ channels (KCa) following ASIC1a-associated intracellular calcium increase, we inhibited KCa *in vivo* by applying apamin and iberiotoxin at the spinal level while WDR neurons were recorded (Fig 5D). Spinal application of these two toxins together had no significant effect on windup. However co-application of apamin and iberiotoxin together with MitTx prevented the strong windup inhibition induced by MitTx alone (Fig 5D–5E). The

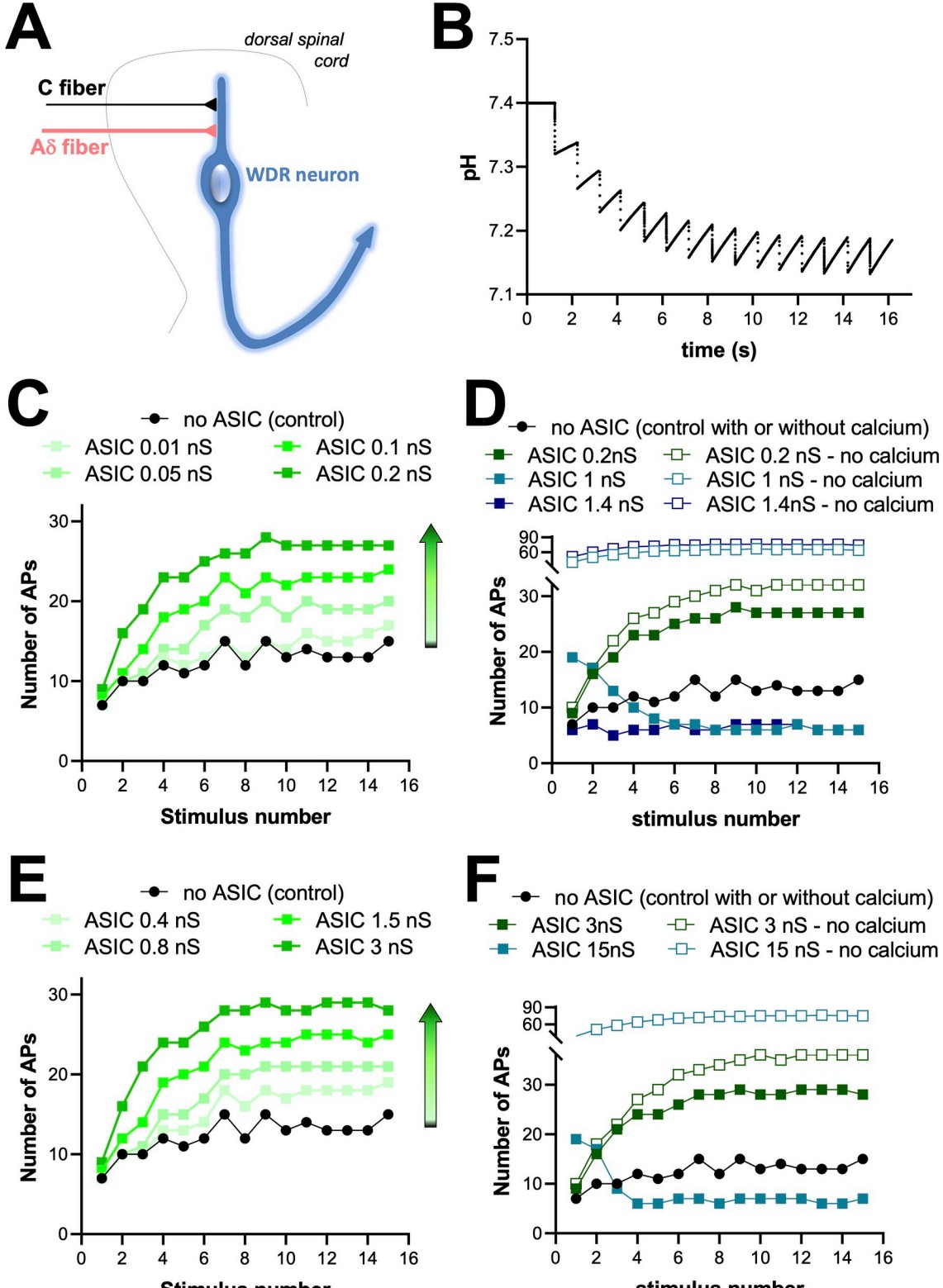

**Fig 4. Computational model of WDR neuron with ASIC1a channel parameters and a synaptic cleft acidification system.** *A*, Schematic representation of the computational model used, with a WDR projection neuron receiving directly a 20-synapse connection from both Aδ-fibers and C-fibers. *B*, Simulation curve showing the evolution of synaptic pH as a function of time during windup-inducing stimulations. *C*, Simulation windup curves obtained without (control, no ASIC, black dots) and with moderate

homomeric ASIC1a conductances ($g$ is 0.01nS, 0.05nS, 0.1nS and 0.2nS, green squares). **D**, Simulation windup curves obtained in control condition (no ASIC, black dots) and with high homomeric ASIC1a conductances (0.2nS, 1nS and 1.4nS, full squares). **E**, Simulation windup curves obtained without (control, no ASIC, black dots) and with moderate heteromeric ASIC1a conductances $g$ is 0.4nS, 0.8nS, 1.5nS and 3nS, green squares). **F**, Simulation windup curves obtained in control condition (no ASIC, black dots) and with high heteromeric ASIC1a conductances (3nS and 15nS, full squares). Compared to control, gradually adding homomeric or heteromeric ASIC1a channels first increases windup, then decreases it. Removing the calcium conductance in ASIC parameters (same conductances, open squares) suppresses the inhibitory effect of high ASIC conductances and strongly potentiates windup.

removal of KCa blockers restored the potency of MitTx to inhibit windup, with an effect that was comparable to the one initially observed, *i.e.*, without pre/co-application of the KCa blockers (Fig 5E).

To summarize, all these data showed that maximal activation of spinal ASIC1a channels by MitTx inhibited windup through a mechanism that is dependent on calcium-activated K⁺ channels. It is in full agreement with the prediction of our mathematical model and further supports both the relevance of the model and the participation of ASIC1a channels to windup in WDR neurons.

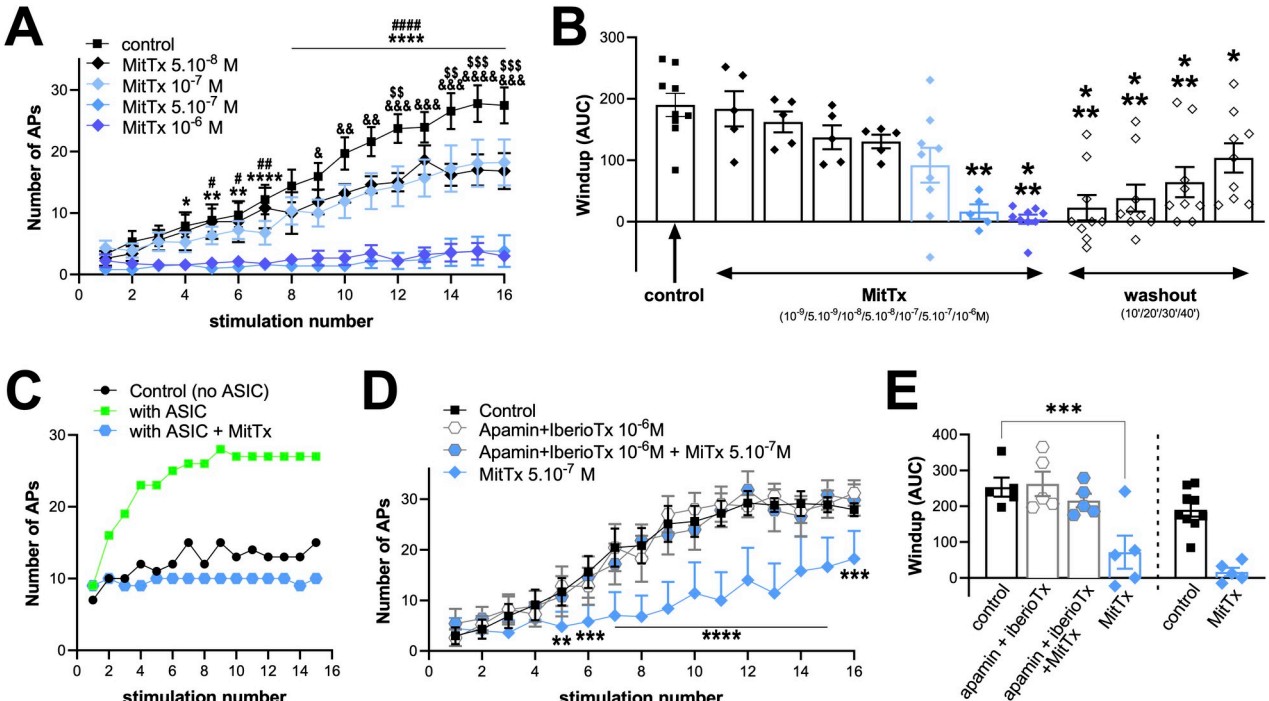

**Fig 5. The ASIC1a activator MitTx inhibits windup *in vivo*. A-B**, Spinal applications of MitTx (2nd) dose dependently and reversibly inhibits windup (n = 5–9; panel A: *$p<0.05$, **$p<0.01$ and ****$p<0.0001$ for control *vs.* MitTx $10^{-6}$ M, #$p<0.05$, ##$p<0.01$ and ####$p<0.0001$ for control *vs.* MitTx $5.10^{-7}$ M, &$p<0.05$, &&$p<0.01$, &&&$p<0.001$ and &&&&$p<0.0001$ for control *vs.* MitTx $10^{-7}$ M, $$$p<0.01$ and $$$$p<0.001$ for control *vs.* MitTx $5.10^{-8}$ M; panel B: *$p<0.05$, **$p<0.01$ and ***$p<0.001$ as compared to control (1st), respectively; Mixed-effect analyses followed by Dunnet's multiple comparison tests). **C**, Simulation of the effect of MitTx on windup (Control: no ASIC conductance; with ASIC: with a native homomeric ASIC1a conductance of 0.2nS; with ASIC+MitTx: sustained full activation (without inactivation) of ASIC1a with a same conductance of 0.2nS). Note that because the ASIC channel is constitutively and fully activated, the outcome of this simulation does not depend on the pH and dynamics of the channel, and is therefore the same for all ASIC types (homomeric, heteromeric). **D-E**, The inhibitory effect of MitTx $5.10^{-7}$M is abolished by spinal pre- and co-application of the KCa blockers apamin and iberiotoxin (D, $10^{-6}$M each). Removing these two blockers restored the windup inhibition induced by MitTx (application sequence: control / Apamin+IberioTx / Apamin+IberioTx+MitTx / MitTx). As a comparison, data already presented in B and showing the effect of MitTx 5.10-7M applied alone, *i.e.*, without pre-application of KCa blockers, are also represented on the bargraph in E (n = 5–9; panel D: **$p<0.01$, ***$p<0.001$ and ****$p<0.0001$ as compared to control, respectively, two-way ANOVA followed by a Dunnet's multiple comparison test; panel E: ***$p<0.001$, one-way ANOVA followed by a Dunnet's multiple comparison test).

## Discussion

The present work describes how ASIC1a channels are involved in the processing of pain message at the level of deep laminae spinal WDR neurons. Spinal ASIC1a activity had already been associated to windup and CFA-induced hypersensitivity of dorsal horn neurons [20]. However, the channel subtypes as well as the subsets of spinal neurons involved remained to be determined. By combining *in vivo* and *ex vivo* electrophysiological recordings, pharmacology and computational modeling, we propose that ASIC1a homomeric channels are functionally expressed in spinal dorsal horn neurons likely corresponding to WDR neurons, where they are basically involved in the windup process and participate to the post-synaptic integration of nociceptive information coming from the periphery. This is in good agreement with the large expression of ASIC1a in spinal cord neurons [19–21]. It is important to mention that other ASIC1a-expressing neurons beside deep WDR neurons, including axons from afferents, could also contribute to the windup modulation since drugs were applied to the entire spinal cord. However, functional characterization of ASIC1a homomers in lanima V WDR neurons strongly supports, together with the WDR computational model, a windup-related role in these neurons.

Post-synaptic expression of ASIC1a channels has been reported in other areas of the central nervous system such as nucleus accumbens and amygdala neurons, where the channels have been involved in synaptic transmission [44,45]. However, expression of different ASIC1a channel subtypes, including homomers and heteromers, remains largely unknown, especially in the different subsets of spinal neurons where they are probably finely tuned at the post- and probably also the pre-synaptic levels. We described here that ASIC1a homomers are at least post-synaptically expressed in deep WDR neurons to participate to windup through an intriguing "bell shape" mechanism. Indeed, inhibiting or maximally activating ASIC channels both lead to the same result, *i.e.*, a windup inhibition. The way inhibitors like mambalgin-1 (or PcTx1) and activators like MitTx inhibit windup is different. If the inhibition of ASIC1a channel subtypes (mambalgin-1 or PcTx1 effects) in WDR neurons produced, as expected, a windup decrease most probably associated with reduced depolarization, the consequence of their over-activation (MitTx effect) is not so straightforward and associated to the calcium conductance of ASIC1a-containing channels. It is important to note that MitTx is a potent activator of ASIC1a channels and provokes an activation that is different from activation of these channels by their natural ligand (*i.e.*, protons). A normal ASIC1a channel activates and inactivates upon acidification, while ASIC1a stays open as long as the MitTx is present because of lack of desensitization and slow reversibility [31]. The amplitude of the activation can also be higher than with the natural ligand. This most probably explains why MitTx, even at low concentrations, has only an inhibitory effect on windup, because it actually mimics situations associated with over-activation of ASIC1a channels instead of their normal activation. Anyway, it strongly argues for an expression in WDR neurons of homomeric ASIC1a channels, which are the only ASICs expressed in the central nervous system described to exhibit a substantial $Ca^{2+}$ permeability (at least in rodents), although our experiments also suggest a $Ca^{2+}$ permeability for ASIC1a/ASIC2a (but not ASIC1a/ASIC2b) heteromers (S2B Fig). It can be hypothesized that, *in vivo*, an excessive synaptic acidification related to intense synaptic activity could, in some conditions, cause an over-activation of ASIC1a channels, ultimately leading to windup inhibition associated to calcium-sensitive $K^+$ channels. This feed-back loop could constitute a protection mechanism that may prevent WDR neurons from over-excitability. Such a protection mechanism could be a more general property of neurons expressing ASIC1a channel subtypes, and especially those having a significant $Ca^{2+}$ conductance like homomeric ASIC1a channels [14].

Our data clearly show that ASICs, contrary to NMDA receptors for instance, are not mandatory to produce windup since their pharmacological inhibition by the toxins PcTx1 or mambalgin-1 decreases, but not abolishes, windup. ASIC1a thus seems to have a rather modulatory effect. The computational model allows to speculate on how ASICs compare with other currents playing a role in windup (S8 Fig). The AMPA and NMDA currents clearly dominate the input to the dendrite, as expected. The dynamic of ASIC currents is rather similar to that of the NK1 receptor associated currents although currents are bigger, which suggest a close contribution of both, at least for low to moderate ASIC conductances. The ASIC currents also have a much bigger amplitude than the calcium-activated non-specific currents iCaAN. The dynamics of the L-type calcium channels, only activating briefly at spike times, is quite different from that of ASICs, which might also explain why the calcium influx from ASICs could inhibit windup although L-type calcium channels participate to windup generation according to Aguiar and colleagues [41]. The small but sustained ASIC current induced by repetitive stimulation appears to be important for the role of ASIC1a in windup, as it may also be the case for its role in LTP that have been reported in different brain areas [44,46,47], although the two processes are different.

Moreover, although the time scale of the acidification shown in Fig 4B is similar to that of windup, it is the intrinsic dynamics of ASICs rather than the time course of acidification that is crucial to the effect of ASICs on windup: as shown in S9 Fig, qualitatively similar results arise for very different acidification time courses. In our simulations, homomeric and heteromeric models operate on slightly different conductance ranges. This is not discriminatory because the exact values of ASIC-associated calcium influx, which are experimentally unknown and arbitrarily chosen in the model, may alter the tipping point of the dual effect of ASICs: the less calcium they let in, the higher the conductance at which ASICs start activating KCa channels and reducing windup (in the high-conductance set-up as in the MiTx/full activation set-up).

The use of computational modeling has been instrumental to the success of our project. Indeed, using one of the very few existing models of WDR neurons and neglecting network effects, we have been able to mathematically reproduce experimentally obtained results, and point towards a biological mechanism realizing predictions, which were further supported by *in vivo* experiments. Such a back and forth exchange between experimental and computational approaches further demonstrates how computational modeling can help understand the complexity of biological events and, in the present case, how unitary mechanisms can be involved in complex physiological processes such as windup. It is noteworthy that this model was able to *(i)* confirm the *in vivo* observations regarding the role of ASIC1a channels in windup, and *(ii)* to predict a windup inhibition upon large activation of ASIC1a channels through calcium-dependent $K^+$ channels, which has then been experimentally validated. The key to these achievements was to modify the model of Aguiar and colleagues [41] by adding different parameters of ASIC1a channels, based on a recently published model [42] and existing data [19], and a synaptic pH model. Several research axes are now possible, including the numerical dissection of the precise mechanisms by which WDR response shows up and is altered by ASIC1a channels, which could provide the operating regime of the WDR neurons and point towards simplified/phenomelogical models of these neurons [48,49]. These simplified models of WDR neurons could also be embedded in a network and help revisit our assumption regarding the role of the network in shaping the WDR response.

Most if not all spinal neurons have been reported to display a functional ASIC1a channel subtype, including homomers and heteromers made of ASIC1a, ASIC2a and ASIC2b subunits [19–21]. Understanding how these different channel subtypes are distributed within the neuronal network of the dorsal spinal cord is an essential point to fully demonstrate how ASICs

participate to spinal pain mechanisms. This work provides both experimental and computational arguments for an original participation of calcium permeable ASIC1a channels in WDR neurons to the pain facilitation process of windup. It also set up an ASIC1a-containing WDR computational model that could be very helpful to further explore the role of ASIC channels in the spinal pain network.

## Material and methods

### Ethics statements

Experimental procedures used in this study were approved by the Institutional Local Ethical Committee (Ciepal-Azur) and authorized by the French Ministry of Research according to the European Union regulations (agreement n°02595.02).

### Animal anesthesia and surgery

Experiments were performed on 5–7 week-old Wistar male rats (Charles River Laboratories) that were housed in a 12 hours light/dark cycle with food and water available *ad libitum*. Animals were anesthetized with isoflurane (Anesteo, France) and placed on a stereotaxic frame (M2E, France) with the head and vertebral column stabilized by ear bars and vertebral clamps, respectively. Limited laminectomies were performed at the T13-L2 segments to expose the dorsal spinal cord and the underlying dura mater was removed.

### *In vivo* recording of dorsal horn neuron activity

Single unit extracellular recordings of lumbar dorsal horn neurons were made using tungsten paralyn-coated electrodes (0.5MΩ impedance, WPI, Europe). The tip of a recording electrode was initially placed at the dorsal surface of the spinal cord using a micromanipulator (M2E, France) and this initial position was set as 0µm on the micromanipulator's micrometer. The electrode was then progressively moved down into the dorsal horn until the receptive field of a spinal neuron was localized on the ipsilateral plantar hindpaw using mechanical stimulations. Two types of mechanical stimulations were used to characterize spinal neurons, including non-noxious brushing and noxious pinching, and we focused on WDR neurons responding to both brush and pinch stimuli. The depth of the WDR neurons selected for this study was >400µm, most probably corresponding to deep neurons from laminae V. Activities of WDR neurons were sampled at 20 kHz, band-pass filtered (0.3–3 kHz) and amplified using a DAM80 amplifier (WPI, Europe), digitized with a A/D-D/A converter (1401 data acquisition system, Cambridge Electronic Design, Cambridge, UK), and recorded on a computer using Spike 2 software (Cambridge Electronic Design, Cambridge, UK).

### Windup protocol and analysis

Once a WDR neuron was isolated, its receptive field was stimulated every 10 minutes with the following protocol: 10 times brushing, to generate Aβ-evoked response, followed by a train of 16 supraliminar electrical repetitive stimulations (1Hz, 4ms pulse width, Dagan S900 stimulator, Mineapolis, USA) to induce windup. Intensity of currents injected for the windup protocols was determined as the intensity required to evoke less than 10 action potentials (APs) at the first stimulation, corresponding to 1.2–3 times the AP thresholds. Controls and drugs were applied consecutively in the same animals for 10 minutes between the stimulation protocols, directly to the dorsal surface of the spinal cord in 40µl of ACSF saline solution that contained (in mM): NaCl 119, KCl 2.5, $NaH_2PO_4$ 1.25, $MgSO_4$ 1.3, $CaCl_2$ 2.5, $NaHCO_3$ 26, glucose 11 and HEPES 10 (pH adjusted to 7.4 with NaOH). Two controls were performed before drug

application, and the average of these controls was used as baseline. APs evoked by WDR were classified according to the time frame at which they were emitted following the electrical stimulation artifact, *i.e.*, 0–20, 20–90 and 90–350 ms for Aβ-, Aδ- and C-inputs, respectively [50]. The remaining APs evoked 350–1,000 ms after the stimulation artifact were classified as the post-discharge activity of WDR neurons. Windup calculations were established by counting the number of APs evoked during the C- and post-discharge parts of recordings. Area under curves (AUC) for windup were calculated from graphs of the number of APs as a function of the stimulus number (windup curves) using Prism software, with the baseline set at the number of APs obtained at the first stimulation for each windup curve.

## Patch-clamp experiments on spinal cord slices

Transverse spinal slices (400μm thick) were prepared from male rats (15–28 days old) as described previously [51]. Rats were deeply anesthetized with urethane (1.9 g/kg, i.p.) and killed by decapitation. The spinal cord was removed by hydraulic extrusion and washed in ice-cold ($\leq$4˚C) sucrose–artificial CSF (ACSF) containing the following (in mm): 248 sucrose, 11 glucose, 26 $NaHCO_3$, 2 KCl, 1.25 $KH_2PO_4$, 2 $CaCl_2$, 1.3 $MgSO_4$ (bubbled with 95% $O_2$ and 5% $CO_2$). The lumbar segment was embedded in 5% agarose, and 400-μm-thick transverse slices were cut with a vibratome (VT1200S; Leica, Germany). Slices were stored at room temperature in a chamber filled with normal ACSF containing the following (in mm): 126 NaCl, 26 $NaHCO_3$, 2.5 KCl, 1.25 $NaH_2PO_4$, 2 $CaCl_2$, 2 $MgCl_2$, 10 glucose (bubbled with 95% $O_2$ and 5% $CO_2$, pH 7.3; 310 mOsm measured). Lamina V putative WDR neurons were recorded based on the localization and the visualization of their large body size. Patch-clamp recordings were obtained with an Axon MultiClamp 200B amplifier coupled to a Digidata 1322A Digitizer (Molecular Devices, CA, USA). Borosilicate patch pipettes (2–4 M$\Omega$) were filled with (composition in mM): 125 KCl, 10 HEPES, 2 $MgCl_2$, 2 MgATP, 0.2 MgGTP (pH 7.3). Neurons were voltage clamped at -80 mV and ASIC currents induced by locally puffing a MES-buffered pH6.6 solution for 5 seconds.

## Patch-Clamp and calcium experiments on HEK293 transfected cells

HEK293 were prepared as described elsewhere [52], before being transfected using JetPEI (Polyplus transfection SA, Illkirch, France) with different plasmids encoding the rat clones of ASIC1a, ASIC2a and/or ASIC2b. Different mixtures of pCI-A1a alone or pCI-A1a + pCI-A2a (1:2 ratio) or pCI-A1a + pCI-A2b (1:2 ratio) with either pIRES2-EGFP (patch-clamp experiments) and/or pIRES2-HcRed (calcium experiments) were used for transfections. Fluorescent cells were then used for patch-clamp or calcium recordings 2–4 days after transfection. For patch-clamp experiments, the whole-cell configuration was used to record membrane currents at a holding potential of -50 mV, and recordings were made at room temperature using an Axopatch 200B amplifier (Axon Instruments) with a 2 kHz low-pass filter and a Digidata 1550 A-D/D-A converter (Axon Instruments) driven by pClamp software (version 11; Axon Instruments), with a sampling rate of 20 kHz. The patch pipettes were made with borosilicate glass capillaries (WPI, France) and were filled with an intracellular solution containing (in mM): 135 KCl, 2 MgCl2, 5 EGTA, and 10 HEPES (pH 7.25 with KOH). The extracellular solution bathing the cells contained (in mM) the following: 145 NaCl, 5 KCl, 2 MgCl2, 2 CaCl2, 10 HEPES (pH 7.4 with N-methyl-D-glucamine), and ASIC currents were induced by shifting one out of eight outlets of a homemade microperfusion system driven by solenoid valves, from a holding control solution at (*i.e.*, pH 7.4 or pH 8.0) to an acidic test solution (pH 5.0 or pH 6.0 or pH6.6 with MES or HEPES as pH buffer). For intracellular calcium measurements, cells were incubated with Fluo4-AM (Invitrogen by Thermo Fisher Scientific) for 45 minutes at

37˚C. Cells were then placed on a wide-field inverted fluorescence microscope (Axiovert200M, Carl Zeiss, Rueil Malmaison, France) equipped with a metal-halide excitation source, an ORCA-Flash4.0 sCMOS camera (Hamamatsu, Massy, France) a YFP filter set (excitation Band Pass 500/20 nm; emission Band Pass 535/30 nm; dichroic mirror Long Pass 515nm) and a PlanApoChromat 63x/1.4 DIC oil immersion objective (pixel size: 100nm/pixel). The fluorescence from HcRed expressing cells was taken at the beginning of each recording using a HcRed filter set (excitation Band Pass 560/40nm; emission Band Pass 630/75nm; dichroic mirror Long Pass 585 nm). Time-lapse Fluo4 images were then taken at 20 Hz in stream mode with Metamorph software V7.10.5 (Molecular Devices), while extracellular medium bathing the cells was rapidly changed for 5 seconds from a pH 7.4 control solution to a pH 6.0 test solution. The Fluo-4 intensity measurements during time were then made in HcRed expressing cells using a homemade ImageJ/Fiji macro-program. HcRed expressing cells were segmented by minimum dark thresholding after a median filtering and their corresponding Region of Interest (ROI) were used to get the mean fluorescent Fluo4 intensity during time. The fluorescence was normalized to the baseline (F/F0) defined for 3 seconds before pH6.0 acidifications.

## Drugs

Synthetic mambalgin-1 was purchased from Synprosis/Provepep (Fuveau, France) and Smartox (Saint Martin d'Hères, France). PcTx1, MitTx and IbTx were purchased from Smartox. Apamin was purified in the laboratory from bee venom. Toxins were prepared as stock solutions in ACSF or extracellular patch clamp solutions, stored at -20˚C, and/or diluted/prepared to the final concentration just before the experiments. For patch-clamp experiments, toxins were applied onto the cells in the control extracellular solution (pH7.4 or pH8.0) containing 0.05% (w/v) BSA for 60 seconds before extracellular acidification. No BSA was used for *in vivo* experiments.

## Statistical analysis

Data are presented as the mean +/- SEM and statistical analysis was performed using Prism software. The statistical tests used to compare different sets of data are indicated in each figure legend.

## WDR neuron mathematical modeling including ASIC1a channel parameters and synaptic cleft acidification

Our computational model of WDR neuron (Figs 4 and S3 and S4) is an elaboration of a model introduced by Aguiar and colleagues [41]. We refer to this work for the model description and only underline our modifications. The model, implemented in NEURON software, was downloaded from ModelDB (https://senselab.med.yale.edu/modeldb/).

By lack of evidence on the polysynaptic nature of the connection between the C-fiber and the WDR neuron, we removed the interneuron from the model in [41] to better fit the experimental spiking dynamics (see Results and S3C and S3D Fig). Our model is thus made of a WDR neuron, with dendrite, soma and axon, which receives a direct input from C-fiber via 20 synapses and from Aδ-fiber via 20 synapses. The Aδ-fiber contacts the WDR neuron with a smaller delay than the C-fiber and its synaptic mechanisms are AMPA and NMDA receptors; further details can be found in the original work [41]. The C-fiber synapses each include AMPA, NMDA and GABA$_A$ receptors with unchanged time dynamics and respective maximum conductances of 6nS, 4nS and 0.3nS, respectively, as well as a neurokinin 1 (NK1) receptor with rise time constant of intermediate value $\tau_{rise} = 150ms$ and maximum conductance of 3pS. In order to finely adjust windup in the model without interneuron (before introducing

**Table 1. Parameters defining the pH-dependence of the steady-state m and h variables for the native homomeric and heteromeric ASIC models.**

|  | $n_m$ | $pH_{0.5m}$ | $n_h$ | $pH_{0.5h}$ | Alpha |
|---|---|---|---|---|---|
| Homomeric native | 1.5 | 6.46 | 4.6 | 7.3 | 1.3 |
| Heteromeric native | 1.94 | 6.03 | 3.82 | 6.74 | 1 |

Note that the work of [19] fit a sigmoid with maximal value alpha = 1.3 (instead of 1) for $h_\infty$, which we respected, having no grounds to create a different model.

ASICs) to experimental data, the calcium-dependent potassium current in the WDR model were tuned to $g_{KCa}$ = 2 $mS \cdot cm^{-2}$ in the soma and $g_{KCa}$ = 2.5 $mS \cdot cm^{-2}$ in the dendrite.

Next, we supplemented our WDR model with a model of ASIC1a channel, with parameters to reproduce either native ASIC1a homomeric or heteromeric currents. The model is a simple Hodgkin–Huxley model of ionic channel, adapted from [42]; some of its parameters were modified to model native currents from [19]. As for the WDR model from [41], we refer to the corresponding publication for the structural and parametric choices that are not explicitly mentioned here. Note however that the null pH shift of 0.15 introduced in their Fig 2C was removed in all models. In these models, the current flowing through the channels is defined as = $g \cdot m \cdot h \cdot (V-E)$, where $g$ is the maximal conductance of the channel, and $m$ and $h$ the activation and inactivation variables, respectively. Both variables are only sensitive to pH.

The native homomeric and heteromeric channel models were built to match the experimental data of respectively "Type 1" and "Type 2" native ASIC current from [19], the most complete analysis of native ASIC channel parameters in spinal neurons to date. The parameters for the asymptotic, pH-dependent values for $m$ and $h$ are given by Eq (1) and Eq (2) with parameters given in Table 1, chosen to fit the values found in [19]:

$$m_\infty = \frac{1}{1 + 10^{n_m \cdot (pH - pH_{0.5m})}}$$ (1)

$$h_\infty = \frac{alpha}{1 + 10^{-n_h \cdot (pH - pH_{0.5h})}}$$ (2)

The order of magnitude of the time constant for the activation variable $m$ is small enough to be considered instantaneous in our experimental setup, and its value as a function of pH was left unchanged from the heterologous ASIC1a model of [42] in all models. The time constant $\tau_h$ of the inactivation variable $h$ was fitted to data of [19].

For the homomeric model, there is little available data in [19]. Therefore several functional forms were fitted to the few available data points, resulting in qualitatively similar results (see S4 Fig). A biased gaussian form was chosen:

$\tau_h = a_1 \, e^{-a_2(pH - b_2)^2} + a_3 \, pH - b_3$, with a$_1$ = 49.196, a$_2$ = 34.632, b$_2$ = 7.144, a$_3$ = 0.95 and b$_3$ = 3.77.

The Gaussian roughly reproduces the shape of $\tau_h$ in the existing heterologous model from [42], but the added affine part was needed to fit the extremal data of [19].

For the heteromeric model, the rates for the relevant pH range (6.6–7.4) were plotted on a logarithmic scale and were found to be roughly aligned. This inspired us to make an exponential fit of the form $\tau_h = ae^{-b(pH - c)} + d$, but, in order to properly account for the time constant being low at both high and low pH values (such as 5), we symmetrized the pH-dependence

with an absolute value. The resulting expression for the time constant is:

$$\tau_h = 42.862 * e^{-5.375*|pH-6.6|} + 1.645 (s).$$

As done in [41] for NMDA and NK1 receptors, we modeled the ASIC1a-dependent increase in intracellular calcium concentration [53,54] by setting 10% of the current entering through ASIC channels (of all models) as calcium current. The maximal ASIC conductance at each synapse varied between 0nS and 3nS for "moderate" conductances (Fig 4C, 4E), thus remaining below that of NMDA receptors, and up to 15nS for the "high" conductances of the heteromeric model (Fig 4F). S6 Fig illustrates the behavior of the two models defined above under standard protocols. Due to experimental evidence suggesting prominent involvement of homomeric ASIC1a channels, all figures and results are given for the homomeric model unless explicitly mentioned.

Based on evidence of the postsynaptic involvement of ASICs in central synaptic transmission [44,45], and of synaptic cleft acidification during transmission [55], we located ASIC channels in the WDR membrane at each synapse with the C-fiber. We modeled synaptic cleft acidification based on the work of Highstein and colleagues [43], with parameters fitted to our needs. The total proton concentration in the synaptic cleft is modeled as:

$$\frac{d}{dt}\left([H^+]\left\{1 + \frac{[B]_0}{[H^+] + K_d}\right\}\right) = q - \frac{[H^+] - [H]_0}{\tau}$$

At every synaptic activation, defined by the presynaptic membrane potential going over -30mV, an input proton current q of 1ms duration is delivered into the cleft, modeling the putative proton sources during synaptic transmission (release of acidic vesicles, proton extrusion by the $Na^+/H^+$ exchanger. . .). The baseline proton concentration is given by the physiological pH as $[H]_0 = 10^{-7.4}M$. The buffer parameters $[B]_0 = 22mM$ and $K_d = 10^{-6.3}M$ were chosen for a model of physiological buffer used by [56]. The values of the proton current q = 0.3mM/ms and time constant $\tau = 0.1ms$, least constrained by experimental data or physiological plausibility, were chosen from a range of acceptable values to adjust the resulting pH range to physiological values and to significantly affect windup (see S4 Fig). The precise choice of q and $\tau$ is not important: S4 and S8 Figs show that values within a certain area yield qualitatively similar results in terms of wind-up and spike counts, although the synaptic acidification undergoes very distinct dynamics.

Fig 4B shows the synaptic cleft acidification during the simulation using the chosen parameters, and S6E Fig shows how the two ASIC models respond to this synaptic acidification.

All simulation protocols were run with 15 stimulations delivered 1s apart starting at 1s, and the distribution of synaptic delays was kept unchanged.

## Supporting information

**S1 Fig. Effect of spinal application of mambalgin-1 and PcTx1 on WDR neuron activities evoked by Aβ and Aδ fibers.** *A,* Typical recordings obtained following stimulation of a WDR neuron receptive field by 10 repetitive brushings (non-noxious stimulations), before (control) and after a 10-min application of mambalgin-1 (30μM) at the spinal cord level. *B,* The total number of AP evoked during the brushing protocol described in A are compared before (control 1 and 2, which represent two brushing experiments that were performed consecutively at 10 min intervals) and after applications of either mambalgin-1 or PcTx1 30μM (n = 11–12, $p<0.05$, one-way ANOVA tests followed by Dunnet's mutiple comparison tests). *C,* Typical recordings showing the activity of a WDR neuron during a windup protocol (16 repetitive electrical stimulations at 1Hz). Only the recordings obtained at stimulation 1, 5, 10 and 15 are

represented. The vertical dashed lines represent the time ranges where the activity of the WDR is considered to be evoked by Aδ or C fibres. **D,** Total number of AP evoked by Aδ during windup protocols before (control) after applications of either mambalgin-1 or PcTx1 (n = 13–14, *, $p<0.01$, paired t test).
(PDF)

**S2 Fig. Characteristics of ASIC1a, ASIC1a/ASIC2a and ASIC1a/ASIC2b channels. *A*,** Peak current ratios (pH6.6/pH5.0) obtained from whole-cell patch-clamp experiments performed in HEK293 cells transfected with either ASIC1a alone, ASIC1a+ASIC2a or ASIC1a+ASIC2b (n = 7–25, One-way ANOVA followed by a Tukey's *post hoc* test with ****p<0.0001). *B*, Fluorescence ratio (F/F0) measured in HEK293 transfected cells loaded with the fluo4 calcium probe following a 5-seconds acidification of the extracellular medium from pH7.4 to pH6.0. The arrow indicates the time at which the pH6.0 acidification was applied (n = 88, 125, 35 and 175 for control, ASIC1a, ASIC1a/ASIC2a and ASIC1a/ASIC2b respectively, Two-way ANOVA followed by a Tukey's *post hoc* test with ***p<0.001 and ****p<0.0001).
(PDF)

**S3 Fig. Adapting the WDR neuron computational model. *A*,** Initial model developed by Aguiar and colleagues [41]. *B*, Using Aguiar's model allows us to reproduce the results on windup when NK1 receptor parameters are removed. *C-D*, Spiking time profiles obtained for the first (stim1) and the fifteenth (stim 15) of the windup protocol (repetitive stimulations at 1Hz). Experimental data (blues curves), representing the mean number of AP per 20ms as a function of time (data from 29–30 neurons), are compared to data predicted by the Aguiar model, with (upper gray marks) or without (upper black marks) the interneuron between C-fiber and WDR neuron.
(PDF)

**S4 Fig. Testing the robustness of the model.** Grid searches for the values of q and $\tau$, the two least constrained parameters of the synaptic cleft pH model, for different models of $\tau_h$. The time constant $\tau$ has no direct physiological meaning and is unconstrained. The proton current q is considered acceptable below the maximum 2mM/ms, which represents the effect of an outward current of 250pA/pF (as was observed in HV1-transfected HEK293 cells depolarized to +90mV in [56]) through a membrane of capacitance 2,4µF/cm$^2$ as measured for spinal cord neurons by [57], into a synaptic cleft of 20nm width. We represent, on a q *vs.* $\tau$ plane, the zones in which *(i)* pH remains within a physiologically plausible range ($\varepsilon$7, red zone), *(ii)* windup matches experimental data ($\varepsilon$ 180, yellow zone) and *(iii)* the number of spikes elicited by the last stimulation matches experimental data ($\varepsilon$22, blue zone). Conditions *(ii)* and *(iii)* together constrain the shape of the windup curve. Black dot represents our chosen model. The maximal ASIC conductance is $\underline{g}$ = 0.2nS for **A, B, C**. The colored lines mark the border of each zone, and the colors resulting from the overlap of several zones are shown in **E**. The red zone represents parameters sets for which the pH remains $\varepsilon$7 over 100 stimulations, in order to exclude parameter choices for which the pH only stays within the physiological range because the simulation is interrupted before pH drops further, rather than because the pH parameters are physiologically valid on the long run. Because the available experimental data for $\tau_h$ in the native homomeric model was insufficient to infer any functional form for $\tau_h$ as a function of pH, different functional forms were tested: (**A**), a Gaussian fit as described in the Methods to reproduce a hump-shaped $\tau_h$ as proposed for the heterologous model [42]; (**B**), an affine form $\tau_h = \max(0, -a\, pH+b)$, with a = 160.4 and b = 1195.32. (**C**), a piecewise-affine form with a maximum at the same pH value as the $m_\infty *h_\infty$ curve,
$\tau_h = \max(0, a_1\, pH - b_1)$ *if* $pH \leq 7.37$ *and* $\tau_h = \max(0, a_2\, pH + b_2)$ *if* $pH \geq 7.37$, with $a_1$

= 10.18, $b_1$ = 49.92, $a_2$ = -558.3 and $b_2$ = 4143). **D**, is a similar grid search for the heteromeric model. The maximal ASIC conductance is $g$ = 3nS (as observed earlier, the heteromeric models acts at higher conductances than the homomeric model). For all three functional forms of the homomeric model, as well as the heteromeric model, the zone in which the parameters respect physiological pH constraints *and* yield windup curves qualitatively matching experimental data, identified by the superposition of all three color zones (brownish-orange), is wide and includes several parameter sets. This suggests both computational and physiological robustness to synaptic cleft pH changes and precise ASIC dynamics. On the one hand, our modelling results do not rely heavily on the choice of the least constrained parameters such as a q, $\tau$ and the function $\tau_h$. On the other hand, a single pH model is valid with different ASIC models, and each ASIC model is valid for a range of pH parameters, suggesting robustness to synaptic heterogeneity and differences of either pH dynamics or ASIC types.
(PDF)

**S5 Fig. Effect of Apamin and Iberitoxin of the windup process.** Part of Fig 5D only showing the effects of Apamin + IberioTx (2nd) compared to control (1st).
(PDF)

**S6 Fig. Behavior of the two types of ASIC models under standard protocols.** Simulation data representing the relative conductance obtained for the recovery from inactivation (**A, C**) and for the pH-dependent activation/inactivation (**B, D**) processes of native ASIC1a homomeric and ASIC1a/ASIC2 heteromeric models, which were elaborated with the type 1 and type 2 parameters, respectively. **E,** Relative conductance of ASIC channels of the two models in response to the synaptic cleft acidification modeled at the synapse (Fig 4B) during the simulation. Notice the different scales associated to the homomeric and heteromeric models, illustrating a difference of the conductance ranges over which the same qualitative behavior is observed. **F**, Relative ASIC conductance of the homomeric and heteromeric models when the channels are activated by a classical pH drop from pH7.4 to pH6.0 (blue) or 7.32 (orange).
(PDF)

**S7 Fig. Effect of varying ASIC maximal conductance on calcium-related currents in the dendrite.** In this figure, we represent the current amplitude of the iCaAN, iKCa and iCa,L currents in the WDR neuron dendrite during the simulation at medium (**A**, **C**; 0.2nS) and high (**B**, **D**; 1.4nS) conductances corresponding respectively to wind-up potentiation by ASICs and wind-up inhibition by ASICs. The dendritic calcium concentration and the membrane potential of the soma and dendrite are also shown. **C** and **D** are close-up views of **A** and **B** respectively, for a better comparison of the iCaAN current values. The higher ASIC conductance clearly correlates with higher calcium concentrations in the dendrite, as well as stronger activation of the hyperpolarizing KCa channels, which we postulate is the mechanism underlying windup inhibition by ASICs. Contrariwise, the increased calcium concentration in the dendrite does not cause much higher activation of the iCaAN channels (which would have the opposite effect). More fluctuations of the membrane potential in C lead to more current fluctuations, but currents fluctuate between roughly the same values the two conditions (i.e., C and D). This is because iCaAN channels are already close to saturation/full opening at the concentrations reached with the moderate ASIC conductance (gating variable m ~ = 0.78–0.92).
(PDF)

**S8 Fig. Comparing ASIC currents to other dendritic currents of the WDR neuron model.** This figure shows the transmembrane currents in the dendrite of the WDR model in order to evaluate how ASICs compare with other currents playing a role in windup. The simulation uses the native homomeric model with maximal conductance 0.2nS. **A**, The ASIC current

(blue) and the sum of all other transmembrane dendritic currents (black) over the whole duration of the simulation. The ASIC current is slowly activating and remains a small contribution compared to the overall current. **B**, **C**, Close-ups of the same graphs with added details of each current: blue curve is the ASIC current, black curve is the total dendritic current, which is the sum of all other shown currents. iKCa, calcium-activated $K^+$ currents. iCa,L, L-type calcium currents. iCaAN, calcium-activated nonspecific cationic currents. The AMPA and NMDA currents, shown in brown, pink, purple and red, clearly dominate, as expected, the input to the dendrite. The ASIC current, shown in blue, is a sustained and increasing current. Its dynamic is rather similar to that of the NK1 receptor associated currents (orange) although the ASIC current amplitude is slightly bigger; this may explain that at low to moderate conductances, ASICs participate to windup as NK1s do (as long as the associated calcium influx is not too big). The ASIC currents also have a much bigger amplitude than the calcium-activated nonspecific currents iCaAN (light blue). The dynamics of the L-type calcium channels (light green), only activating briefly at spike times, is quite different from that of ASICs, which might also explain why the calcium influx from ASICs may inhibit windup although L-type calcium channels participate to windup generation according to Aguiar *et al*. [41].
(PDF)

**S9 Fig. The contribution of ASICs to windup does not rely on the dynamics of our chosen pH model.** This figure reproduces the results of Fig 4B, 4C and 4D for other choices of the pH parameters q and yielding very different synaptic cleft pH dynamics. **A, E**, evolution of the synaptic cleft pH for two other sets of parameters (compared to Fig 4B) in the validity zone defined by S4 Fig: the alternative 1 parameter set is (q = 1mM/ms, τ = 0.01ms) which results in a very fast acidification time course; the alternative 2 parameter set is (q = 0.05mM/ms, τ = 1ms) which contrariwise results in a very slow acidification time course. **B, F**, reproduction of Fig 4C for alternative 1 and alternative 2 parameter sets respectively: progressively increasing the ASIC1a maximal conductance potentiates windup, or, from the experimental point of view, inhibiting ASIC1a channels reduces windup. **C, G**, reproduction of Fig 4D for alternative 1 and alternative 2 parameter sets respectively: higher ASIC maximal conductances inhibit windup, but only if ASICs are permeable to calcium. **D, H,** ASIC conductance and current during the windup protocol with alternative 1 and alternative 2 parameter sets respectively for maximal conductance 0.2nS. The time course of the ASIC currents depends on the set of parameters, but not the time course of the windup. Overall, these qualitatively similar results with drastically differing pH time courses show that the effect of ASIC1a channels on windup is brought on by their own intrinsic dynamics rather than by the dynamics of the synaptic cleft acidification. Synaptic cleft acidification is required to activate ASICs, but the resulting windup follows its own time scale, independent of the time course of the acidification.
(PDF)

## Acknowledgments

We thank Drs A. Baron, S. Diochot, J. Noel, and M. Salinas for helpful discussions, V. Friend and J. Salvi-Leyral for technical support, and V. Berthieux for secretarial assistance. We also thank the Centre National de la Recherche Scientifique, the Institut National de la Santé et de la Recherche Médicale and the University Côte d'Azur for their supports.

## Author Contributions

**Conceptualization:** Magda Chafaï, Ariane Delrocq, Perrine Inquimbert, Eric Lingueglia, Romain Veltz, Emmanuel Deval.

**Data curation:** Magda Chafaï, Ariane Delrocq, Perrine Inquimbert, Ludivine Pidoux, Kevin Delanoe, Maurizio Toft, Emmanuel Deval.

**Formal analysis:** Magda Chafaï, Ariane Delrocq, Ludivine Pidoux, Kevin Delanoe, Maurizio Toft, Romain Veltz, Emmanuel Deval.

**Funding acquisition:** Ludivine Pidoux, Eric Lingueglia, Romain Veltz, Emmanuel Deval.

**Investigation:** Magda Chafaï, Ariane Delrocq, Perrine Inquimbert, Ludivine Pidoux, Kevin Delanoe, Maurizio Toft, Emmanuel Deval.

**Methodology:** Magda Chafaï, Ariane Delrocq, Frederic Brau, Romain Veltz, Emmanuel Deval.

**Project administration:** Romain Veltz, Emmanuel Deval.

**Resources:** Frederic Brau, Eric Lingueglia, Romain Veltz, Emmanuel Deval.

**Software:** Ariane Delrocq, Romain Veltz.

**Supervision:** Romain Veltz, Emmanuel Deval.

**Validation:** Magda Chafaï, Ariane Delrocq, Perrine Inquimbert, Ludivine Pidoux, Kevin Delanoe, Maurizio Toft, Romain Veltz, Emmanuel Deval.

**Writing – original draft:** Ariane Delrocq, Romain Veltz, Emmanuel Deval.

**Writing – review & editing:** Magda Chafaï, Ariane Delrocq, Perrine Inquimbert, Frederic Brau, Eric Lingueglia, Romain Veltz, Emmanuel Deval.

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
