## [Decision Letter · Decision Letter 0]

21 Dec 2021

Dear Dr. Deval,

Thank you very much for submitting your manuscript "Involvement of ASIC1a channels in the spinal processing of pain information by deep projection neurons" for consideration at PLOS Computational Biology.

As with all papers reviewed by the journal, your manuscript was reviewed by members of the editorial board and by several independent reviewers. In light of the reviews (below this email), we would like to invite the resubmission of a significantly-revised version that takes into account the reviewers' comments.

We are pleased to see reviewers pointing out the importance of your work and findings of interest. Reviewers further provide extensive suggestions on how your manuscript can be approved and we feel these will strengthen your work. We look forward to your updated manuscript.

We cannot make any decision about publication until we have seen the revised manuscript and your response to the reviewers' comments. Your revised manuscript is also likely to be sent to reviewers for further evaluation.

Sincerely,

Erik Fransen, Ph.D.

Guest Editor

PLOS Computational Biology

Kim Blackwell

Deputy Editor

PLOS Computational Biology

Reviewer's Responses to Questions

**Comments to the Authors:**

Reviewer #1: This study investigates the contribution of ASICs to a neuronal plasticity mechanism in the spinal cord, termed windup, that leads to chronic pain. The authors show first with in vivo experiments that the windup is partially inhibited by application of each of two different ASIC inhibitors, Mambalgin-1 or PcTx1. Based on a patch-clamp characterization of neurons of lamina V of the dorsal horn, they propose that the ASICs in these cells are mostly heteromeric channels containing ASIC1a and ASIC2 subunits. The authors present then a computational model of WDR neurons, which can produce windup that depends on ASIC function. This model shows that at high ASIC current densities, windup can be inhibited. This is consistent with the effect of the ASIC activator MitTx that inhibits the development of a windup. Experimental inhibition of Ca-activated K channels prevents this effect of Mit-Tx. The authors suggest that ASICs contribute in a biphasic way to WDR neuron signaling, with potentiation at low, and inhibition at high ASIC activities.

Major points

1. Based on the pharmacology of the windup, the functional characterization of ASIC currents in dorsal horn neurons, and the kinetic models, the authors conclude that ASIC1a/ASIC2 heteromeric ASICs are responsible for the windup. The basis for this conclusion is however very weak. It contradicts the observation by the authors that windup is inhibited by PcTx1, which inhibits only ASIC1a homomeric channels at the experimental conditions used (see PMID 27277303). The functional characterization of ASIC currents in the neurons of lamina V shows inhibition by PcTx1, identifying them as mostly ASIC1a homomer-mediated currents. This contradicts the conclusion made. Regarding the experiments shown in Fig. 2, it is indicated that the conditioning pH was 7.4; in the methods it is said that the slices were kept at pH7.3. Was the measuring solution different from this storage solution? The authors should determine experimentally the pH dependence of the currents, since this parameter is very different between ASIC1a homotrimers and ASIC1a-2 heterotrimers and would allow an experimental distinction.

2. In the model, a calcium permeability is attributed to heteromeric ASICs. The general view in the field is that only ASIC1a homotrimers are calcium permeable. Only one single study showed a calcium permeability of ASIC1a-ASIC2b channels. The authors should provide experimental proof of such a calcium permeability. In a heterologous system these would be straightforward experiments.

3. More information about the model must be provided to allow the reader to better understand it and evaluate its quality. This should include description of the behavior of the two ASIC types in standard protocols, such as pH dependence of activation, inactivation, kinetics of inactivation, recovery from inactivation. The currents generated in the two ASIC models by the pH change shown in Fig. 3B should be shown. From their characterization of lamina V neurons in Fig. 2, the authors should estimate the current density of ASICs in these neurons; they should discuss how this value correlates with the different values that they used in Fig. 3C. Clarify also in the legends whether the conductance values shown in Fig. 3 correspond to the maximal peak conductance that would be reached at the most acidic pH. In the description of the model in the methods, no properties of the A-delta fiber synapses are indicated. Spell out the name of the "NK1 receptor" and provide a reference. On p.19 in the methods, it is said that the ASIC1a homomeric model was unable to reproduce the experimental results. It would be good to indicate, due to which differences between ASIC1a homomers and heteromers the ASIC1a model could not produce a windup.

The model of acidification of the synaptic cleft appears very hypothetical and based more on ASIC function than on information about pH changes. The amplitude of the pH change indicated in Fig. 3B appears to be much bigger than expected from most studies that investigated such pH changes. The authors should provide support from the literature for these predicted pH changes.

4. The most important findings of this study are experimental. I think that an experimental journal would be better adapted to the content of the manuscript.

Specific points

1. Contribution of calcium entry to the inhibition of windup at high ASIC activity. Could the authors determine with their models the time course of calcium entry in the different conditions used in Figs. 3C-D and 4C, and evaluate how the inhibition of the windup correlates with calcium entry? It should also be tested and shown in the model, whether for the induction of windup a calcium permeability of the ASICs is required.

2. p.5, upper part of the page, the description of the actions of PcTx1 and Mambalgin is somewhat misleading; PcTx1 inhibits heteromers only under very specific conditions that are different from the ones used in this study. This should be clarified.

3. Regarding the windup protocols, were different conditions (as e.g. control and different drugs) tested in the same animals? If yes, this should be mentioned in the methods, and it should be indicated how long the intervals between different tests were.

4. p.9, 2nd paragraph, regarding Fig. 4B it is said that the effect of MitTx is fully reversible. However, the mean value after washout is probably close to 60% of the control value, thus this statement should be corrected.

5. p15, windup protocol, clarify to what the AUC measured in the windup protocols corresponds. Is it the AUC calculated from the graph of the number of APs as a function of the stimulus number?

6. Figure 1B, according to the text, the 16 stimulations in the windup protocol are applied at 1 Hz. In the figure however, the interval between the spikes does not correspond to 1s.

7. For the toxin solutions, was there BSA used? Was BSA also included in the control conditions?

8. In Fig. 2A, 3 control applications of pH6.6 were recorded, plus 5 applications (applications 4-8) during the administration of the toxin. Control experiments should be shown in which applications 4-8 are recorded in the absence of any drug.

9. Could you also describe the experimental conditions for ASIC1a-expressing HEK cells? Indicate also whether the recombinant ASIC1a was from rat.

10. Indicate in the figure legends the timing of drug applications; indicate whether they were applied only during the windup protocols, or if there was a pre-application, how long this was.

11. Supplementary Figure 1, indicate exactly what is meant with "control 1 and 2". Should the PcTx1 concentration in the legend rather be 30 nM? I am somewhat lost in panel C. I understand that stimulations of the windup protocol are applied once per second. However, the traces shown in panel C are much longer than 1s.

12. The supplementary figure is very difficult to understand. There are 3 conditions mentioned in the legend, however only 2 of them have an attributed color. In the graphs, there are colored lines and colored surfaces. There is a purple surface whose meaning is not defined.

Reviewer #2: The manuscript by Chafai et al provides functional evidence for a role of the ASIC1a subunit in the process of “wind-up” in WDR neurons in lamina V of the spinal dorsal horn in rats. Wind-up measured as a progressive increase in evoked activity in WDR neurons was induced with electrical stimulation at C-fibre intensity. Wind-up was reduced by ASIC inhibition with PcTx-1 and Mambalgin-1. Whole cell recordings from large diameter neurons in lamina V showed inward currents in response to pH challenge to 6.6 that were blocked by both PcTx-1 and Mambalgin-1. Numerical simulations indicate that a progressive fall in pH leads to an increase in an ASIC conductance that can produce firing patterns similar to those observed with wind-up. Interestingly, application of MitTx, an activator of ASIC1a, also reduced wind-up. To reconcile this apparent paradox, a permeability of ASIC1a to calcium was postulated to lead to activation of calcium-activated potassium channels and thus a reduction in wind-up. A role for calcium activated potassium channels was confirmed by experiment and simulation.

This is an interesting manuscript that provides a functional characterization of putative ASIC currents in WDR neurons. Pharmacological blockade of ASIC reduced wind-up and this experimental result was consistent with results from numerical simulations using an ASIC conductance. In contrast, pharmacological activation of ASIC also reduced wind-up and to explain this, the authors postulate calcium entry via ASIC1a heteromers and ensuing hyperpolarization via IK and BK. This second dataset requires additional work to substantiate the involvement of calcium as outlined below.

The Figures present the data clearly and the manuscript is generally well written.

Major points.

1. There is a striking dichotomy in that both blockade and activation of ASIC reduce wind-up. The authors recognise this point and take up an explanation in the second paragraph of the discussion. However, the explanation provided here is ambiguous and conflates several ideas (outlined below). From my perspective to clarify the role of calcium additional experimental evidence is required to support the idea of calcium entry via ASIC in WDR neurons.

To re-iterate, toxin blockade of ASIC reduced wind-up. This implies that ASIC contributes to the wind-up process in the following manner. An ASIC conductance is low at rest and then during wind-up, as synaptic pH falls, ASIC is progressively recruited to enhance the wind-up process. The toxin blockers PcTx-1 and Mambalgin-1 both prevent this.

However, toxin activation of ASIC with MitTx also reduced wind-up. Assuming the same starting conditions, with a low ASIC conductance at rest, channel opening with MitTx should initially enhance wind-up. It does not. Rather, the clear result from the concentration response profile in Figure 4A&B is that MitTx either did not affect or reduced wind-up.

This is an intriguing result and since sits at the crux of the manuscript.

To explain reduced wind-up with ASIC activation by MitTx, the authors postulate that MitTx opens an ASIC conductance permeable to calcium. In the same sentence however, the idea of maximal channel activation is brought up by stating that “the consequence of their over-activation (MitTx effect) was unexpected and associated to the calcium conductance of ASIC1a-containing channels. (pp.11, last 3 lines). Two concepts are conflated here, specifically (1) maximal channel opening and (2) calcium permeability of ASIC. Taking these two ideas in turn. Firstly, maximal activation leads to calcium entry. The problem here is that the concentration-response profile for MitTx shown in Figure 4 demonstrates that MitTx never potentiates wind-up. This would imply that MitTx, even at low concentrations, maximally opens ASIC and this seems to be unlikely. To be specific, how does the ASIC at rest get to the maximally activated and thus calcium permeable state without showing some evidence of increasing the Na influx assigned to it during the process of wind-up.?? In addition, why does the calcium permeability not manifest during electrically evoked wind-up, i.e. without toxins under normal circumstances?

The second idea of calcium permeability is not clearly formulated in the text. My interpretation is that there are multiple sub-types of ASIC channels in WDR neurons and this is likely. In which case, MitTx activates an ASIC channel population comprising splice variants such as ASIC1a/ASIC2a that have a predominant Ca-conductance. In contrast, the ASIC channels recruited by electrical stimuli (and blocked by PicTx and Mambalgin-1) represents an ASIC population with low Ca-permeability – is this the case? The discussion does not adequately clarify this. Irrespective of the interpretation I would like to see experimental evidence demonstrating an increase in calcium in WDR neurons, at least in response to MitTx and better yet a smaller or absence of calcium increase during wind-up to electrical stimulation. This could be done using a calcium reporter. One might expect that during electrical wind-up the calcium increase is modest while in response to MitTx a prominent calcium increase ensues. Increases in calcium via calcium channel activators or perhaps thapsigargin mediated store release might also be valuable in this context.

2. Although it is stated in the text on pp.9 (second to last sentence) that apamin and iberotoxin in combination do not affect wind-up - it would be important to show this data as part of a Figure.

3. Drugs were applied to the entire spinal cord and thus any number of neurons, including axons from afferents are equally likely to be affected. The question then is, to what extent is the reduction in wind-up with both PcTx-1 and Mambalgin-1 not due to blockade of WDR neurons directly but rather due, at least in part, to blockade of primary afferent signals? The authors point out that both toxins reduced the response to the first stimulus and acknowledge that this represents a reduction in primary afferent input. How can one dissociate at which site the reduction in wind-up is taking place? and can this be done in a quantitative manner, eg. 40% reduction in primary afferent input and 20% from WDR?? This seems to be a major shortcoming of the manuscript.

4. The summary figures show paired data with values before and after drug application (eg. 1C&Finsets; 1D,E,G&H; 2D; 4E; Supp1AB&D). To better visualise the paired nature of these datasets, it would be helpful to join pairs of dots with a line. Showing independent clouds of data points is misleading for paired data.

Minor points:

1. Is the reference to Liu et al (2018) cited in the discussion pp.11 (beginning on last line) “This latter point strongly argues for an expression in WDR neurons of heteromeric ASIC1a/ASIC2b channels, which have a significant Ca2+ permeability [38].” Correct because Liu et al (2018) does not show that ASIC1a/ASIC2b channels have an elevated Ca-permeability.

2. The methods state on pp. 15 (Windup protocoland analysis paragraph, last sentence) ”Area under curves (AUC) for windup curves were determined using Prism software.” and I assume this means simply the integral of AP counts over time. Is this correct? and could this procedure be explained in more detail?

3. The species should be mentioned earlier, either in the title or the abstract.

4. Supplementary Figure 1, panel C shows prominent 50Hz noise in the signal – would it be possible to present a cleaner recording.

Reviewer #3: Windup is an important pain facilitation process in the spinal dorsal horn, characterized by a progressive, frequency-dependent increase in the excitability of WDR nociceptive neurons to repetitive stimulation of primary afferent nociceptive C-fibers. Windup plays an important role in spinal neuronal plasticity and is of unquestionable interest to understand pain processing in the dorsal horn. As the authors point out, the exact mechanism(s) behind windup is(are) still not fully understood.

Throughout the years (windup was originally reported ~half a century ago), different groups have been revealing different contributors for this facilitation process. Not so many groups have taken an integrative approach providing a holistic analysis where the effective role of different contributors are weighted. It should be noted that, as 1 Hz is the typical stimulation frequency in windup protocols, any mechanism with a “recovery” time constant/scale larger than 1 sec will potentially contribute to signal integration and an observable effect on windup profiles.

In this work the authors point to yet another contributor for action potentials windup in WDR neurons: acid sensing ion channels.

This is a clear and well-written manuscript combining in silico, ex vivo and in vivo experiments to verify and assess the contribution of ASIC currents is action potentials windup in WDR neurons.

Questions

1) The choice of the dynamics for the synaptic cleft H+ concentration is critical in shaping the windup profile. How is the time scale of the acidification “windup” (Fig. 3B) related with the time scale of the windup in the number of APs? In other words, is the APs windup profile (time constant) the same as the pH profile (time constant)?

2) No need for a sensitivity analysis on the new ASIC model + ProstProtCleftDyn model, but some insight would be relevant/useful on how robust the conclusions are regarding changes in the parameters of these new models.

3) Regarding this point “This study demonstrates a possible dual contribution to windup of calcium permeable ASIC1a/ASIC2 channels in deep laminae projecting neurons, promoting it upon moderate channel activity, but ultimately leading to calcium-dependent windup inhibition associated to potassium channels when activity increases”. As far as I understood, your model includes both Ca-dependent potassium currents and non-specific current dependent on intracellular Ca concentration (iCaAN). According to Fig 3D, iCaAN in your model does not seem to be needed to provide the temporal integration supporting windup. If iCaAN is not needed, I would state that.

4) I would argue that “total number of AP” nor “area under curves (AUC)” are the best metrics to characterize (and compare) windup curves. The ratio between the (asymptotic) maximum number of APs and the number of APs in response to the first stimulation seems to be a better metric. It is more robust for comparisons, especially in curve comparisons were the number of APs in the first response is different.

5) For the sake of advancing the understanding on the facilitation process of windup, it would be important to have in the Discussion more information on how ASICs compares with the other currents playing a role in windup (e.g. NMDA receptors, NK1 receptors, and L-type calcium channels). Are ASICs necessary and/or sufficient to produce APs windup? This should be discussed taking into account a potential therapeutic strategy based on ASICs inhibition.

Minor issues:

- Inconsistent use of “Fig” and “Figure”

- “was decrease by 20% in the presence of mambalgin-1” -> “decreased”

- When running the NEURON simulation (using mosinit.hoc file), the title of the window should be corrected (it retains the name of a previous publication).

- I would suggest that mosinit.hoc would already produce a figure with the .dat file. But this is just a suggestion thinking on a colleague that, for a quick test, may want to run the simulation without a NEURON+Python installation

**Have the authors made all data and (if applicable) computational code underlying the findings in their manuscript fully available?**

Reviewer #1: Yes

Reviewer #2: Yes

Reviewer #3: None

PLOS authors have the option to publish the peer review history of their article (what does this mean?). If published, this will include your full peer review and any attached files.

Reviewer #1: No

Reviewer #2: No

Reviewer #3: No
---

## [Decision Letter · Decision Letter 1]

22 Dec 2022

Dear Dr. DEVAL,

Thank you very much for submitting your manuscript "Dual contribution of ASIC1a channels in the spinal processing of pain information by deep projection neurons revealed by computational modeling" for consideration at PLOS Computational Biology. As with all papers reviewed by the journal, your manuscript was reviewed by members of the editorial board and by several independent reviewers. The reviewers appreciated the attention to an important topic. Based on the reviews, we are likely to accept this manuscript for publication, providing that you modify the manuscript according to the review recommendations.

I'm pleased to inform you that your updated manuscript was received very well by the reviewers. I would only like you to make a few updates to the text following the comments of reviewer 1. After your brief update, I will be able to accept your manuscript for publication.

Sincerely,

Erik Fransen, Ph.D.

Guest Editor

PLOS Computational Biology

Kim Blackwell

Section Editor

PLOS Computational Biology

I'm pleased to inform you that your updated manuscript was received very well by the reviewers. I would only like you to make a few updates to the text following the comments of reviewer 1. After your brief update, I will be able to accept your manuscript for publication.

Reviewer's Responses to Questions

**Comments to the Authors:**

Reviewer #1: The authors have provided additional information and data and have responded to most of my comments. There are a few points that should be addressed:

1. In Fig. S6E, the relative conductance m*h under the synaptic cleft acidification pattern is displayed. These conductances seem low and extremely low for the homomeric and heteromeric models, respectively. To allow the reader to have a point of comparison, the authors need to indicate (in the text or the legend) the m*h value that is reached at the peak following a typical ASIC1a-activating pH change, such as from pH7.4 to pH6.0 for the ASIC1a homotrimer and for the ASIC1a-containing heteromer model.

2. Fig. S7 is very useful for the understanding of the model. Based on this figure it is concluded that the higher ASIC conductance in B/D does not change much the ICaAN. The upper panel of C is however almost completely red and green, very different from the corresponding panel in D. Does the high color content in C indicate that ICaAN and ICa,L are fluctuating at a high frequency between different current amplitudes, while in condition D, ICaAN reaches soon after each spike a maximal inward current amplitude, from which it decreases slowly until the next spike? And ICa,L would be active only during the spike? The quality/resolution of this figures is not good enough to see any details. The authors need to better describe and interpret this model behavior in the text of the manuscript.

The plots of Ca concentrations are the same in A and C, and B and D. It is not necessary to show them twice. However, it is important to show the membrane potential under these protocols, since the windup is determined as a change in the number of action potentials after each electrical stimulation. This could be done, for the two conditions, in either Fig. S7 or S8, over the whole time of a windup protocol.

3. The resolution of most supplementary figures is very low. This may be due to the merging of several files for review. The authors should make sure that the uploaded supplementary data files are of sufficient quality.

4. p10, line 245, "Inhibition of windup by MitTx was almost reversible....", more correct would be "Inhibition of windup by MitTx was partially reversible....".

5. Fig. S9, the unit in the x-axis label should be s, not ms. In this figure, the resulting ASIC currents induced by the two pH patterns of A and D should be shown.

6. Overall it seems that under the conditions of the windup protocol, the ASIC currents are very low but still contribute importantly to the windup. This aspect should be discussed. Could there be any parallels between this function and the role of ASIC activity in LTP, where ASIC currents are also very small?

Reviewer #2: No further comments

Reviewer #3: The revised manuscript incorporated the recommendations and clarifies the points that were raised. The additional (supplementary) figures complement very well the information in the manuscript.

This is an interesting/relevant study properly combining computational and experimental approaches to assess the contribution of ASIC currents is action potentials windup in WDR neurons.

No additional revisions are recommended.

**Have the authors made all data and (if applicable) computational code underlying the findings in their manuscript fully available?**

Reviewer #1: Yes

Reviewer #2: Yes

Reviewer #3: None

PLOS authors have the option to publish the peer review history of their article (what does this mean?). If published, this will include your full peer review and any attached files.

Reviewer #1: No

Reviewer #2: No

Reviewer #3: No

Figure Files:

Data Requirements:

Reproducibility:

References:

---

## [Editor Report · Decision Letter 2]

3 Mar 2023

Dear Dr. DEVAL,

We are pleased to inform you that your manuscript 'Dual contribution of ASIC1a channels in the spinal processing of pain information by deep projection neurons revealed by computational modeling' has been provisionally accepted for publication in PLOS Computational Biology.

Best regards,

Daniele Marinazzo

Section Editor

PLOS Computational Biology

Kim Blackwell

Academic Editor

PLOS Computational Biology

---

## [Editor Report · Acceptance letter]

29 Mar 2023

PCOMPBIOL-D-21-01920R2 

Dual contribution of ASIC1a channels in the spinal processing of pain information by deep projection neurons revealed by computational modeling

Dear Dr DEVAL,

I am pleased to inform you that your manuscript has been formally accepted for publication in PLOS Computational Biology. Your manuscript is now with our production department and you will be notified of the publication date in due course.

With kind regards,

Zsofia Freund
